# Peripheral and lung resident memory T cell responses against SARS-CoV-2

Judith Grau-Expósito [1,9], Nerea Sánchez-Gaona[1,9], Núria Massana[1], Marina Suppi [1], Antonio Astorga-Gamaza[1], David Perea[1], Joel Rosado[2], Anna Falcó[1], Cristina Kirkegaard[1], Ariadna Torrella [1], Bibiana Planas[1], Jordi Navarro [1], Paula Suanzes [1], Daniel Álvarez-Sierra [3], Alfonso Ayora[4], Irene Sansano[5,6], Juliana Esperalba[7], Cristina Andrés [7], Andrés Antón [7], Santiago Ramón y Cajal [5,6], Benito Almirante[1], Ricardo Pujol-Borrell [3,8], Vicenç Falcó [1], Joaquín Burgos [1], María J. Buzón [1,10]✉ & Meritxell Genescà [1,10,10]✉

Resident memory T cells (T_RM) positioned within the respiratory tract are probably required to limit SARS-CoV-2 spread and COVID-19. Importantly, T_RM are mostly non-recirculating, which reduces the window of opportunity to examine these cells in the blood as they move to the lung parenchyma. Here, we identify circulating virus-specific T cell responses during acute infection with functional, migratory and apoptotic patterns modulated by viral proteins and associated with clinical outcome. Disease severity is associated predominantly with IFNγ and IL-4 responses, increased responses against S peptides and apoptosis, whereas non-hospitalized patients have increased IL-12p70 levels, degranulation in response to N peptides and SARS-CoV-2-specific CCR7⁺ T cells secreting IL-10. In convalescent patients, lung-T_RM are frequently detected even 10 months after initial infection, in which contemporaneous blood does not reflect tissue-resident profiles. Our study highlights a balanced anti-inflammatory antiviral response associated with a better outcome and persisting T_RM cells as important for future protection against SARS-CoV-2 infection.

[1] Infectious Diseases Department, Vall d'Hebron Institut de Recerca (VHIR), Vall d'Hebron Hospital Universitari, Vall d'Hebron Barcelona Hospital Campus, Barcelona, Spain. [2] Thoracic Surgery and Lung Transplantation Department, Vall d'Hebron Institut de Recerca (VHIR), Vall d'Hebron Hospital Universitari, Vall d'Hebron Barcelona Hospital Campus, Barcelona, Spain. [3] Diagnostic Immunology Group, Vall d'Hebron Institut de Recerca (VHIR), Vall d'Hebron Hospital Universitari, Vall d'Hebron Barcelona Hospital Campus, Barcelona, Spain. [4] Occupational Risk Prevention Unit, Vall d'Hebron Hospital Universitari, Vall d'Hebron Barcelona Hospital Campus, Barcelona, Spain. [5] Pathology Department, Vall d'Hebron Hospital Universitari, Vall d'Hebron Barcelona Hospital Campus, Barcelona, Spain. [6] Departament de Ciències morfològiques, Universitat Autònoma de Barcelona, Universitat Autònoma de Barcelona, Bellaterra, Spain. [7] Respiratory Viruses Unit, Microbiology Department, Vall d'Hebron Institut de Recerca (VHIR), Vall d'Hebron Hospital Universitari, Vall d'Hebron Barcelona Hospital Campus, Barcelona, Spain. [8] FOCIS Center of Excellence, Vall d'Hebron Hospital Universitari, Vall d'Hebron Barcelona Hospital Campus, Barcelona, Spain. [9] These authors contributed equally: Judith Grau-Expósito, Nerea Sánchez-Gaona [10] The authors jointly supervised this work: María J. Buzón, Meritxell Genescà ✉email: mariajose.buzon@vhir.org; meritxell.genesca@vhir.org

The outbreak of a novel coronavirus SARS-CoV-2 led to a global health emergency, the COVID-19 pandemic. While a great effort has been focused on vaccine development, many questions remain unanswered that are necessary to properly manage patients and inform vaccine assessment. To this end, identifying the development of a protective immune response after natural infection and characterizing the correlates of protection would greatly inform on the best strategy to stimulate a protective response by immunization. Moreover, identifying specific immunological parameters capable of predicting disease control (i.e., no hospitalization) could provide new biomarkers to support medical decisions for patients. Most efforts to measure or induce immunity rely on neutralizing antibodies, which can certainly limit infection; however, antibody detection is not only inconsistent in infected or convalescent patients[1–4] but may also wane with time as shown for other coronaviruses[5,6], although their stability may also depend on the antigen targeted[7].

Virus-specific T cells against SARS-CoV-2 have been shown to develop against coronaviruses[4,8–14] and specific memory T cells persisted in SARS-recovered patients for up to 6 years post-infection[5]. Thus, T cells may potentially provide long-term immunity as demonstrated for other viral infections such as SARS or influenza[15–18]. In this sense, mouse models of SARS-CoV-1 infection demonstrated that both CD8+T and CD4+T cells are critical for viral clearance[15,17]. Considering that SARS-CoV-2 interacts with the host at the respiratory tract mucosal interface, T cells strategically positioned within these surfaces, may be essential to limiting infection. A key role for resident memory T cells (T$_{RM}$) in protection against pathogen challenge has been established for many tissues, including the lung[5,17,19–22]. These cells, which are strategically located both in the lung airways and in lung interstitial tissue, include CD4+ and CD8+T cells designed to limit re-infections locally. In the context of respiratory infection models such as influenza, CD8+T$_{RM}$ have shown to confer cross-protection against different strains[22], and both influenza-specific CD4+ and CD8+T$_{RM}$ have been identified[16,23]. Importantly, optimal protection against SARS-CoV-infected mice was conferred by airway memory CD4+T cells secreting both, proinflammatory interferon (IFN) γ and anti-inflammatory interleukin (IL)-10 molecules[17]. Thus, a broader spectrum of T helper (Th) profiles should be included when addressing virus-specific T cells. Further, recruitment of these cells from circulation may depend on the expression of molecules such as chemokine C-X-C motif receptor (CXCR) 3, which, besides mediating chemotaxis toward inflamed tissue of a biased Th1 profile, appears critical for the recruitment of pulmonary T cells that control infection[18,24,25]. While such recruitment could partially contribute to the decrease of circulating lymphocytes and thus favor tissue infiltration, many other factors may explain the observed lymphopenia associated with COVID-19 disease severity[26–29]. In this regard, increased susceptibility of both specific and bystander CD4+ and CD8+T lymphocytes to apoptotic cell death has been observed in other viral infections[30–32], which could be linked to increased glycolysis[33,34].

Here, we address several important questions related to early control of SARS-CoV-2 infection mediated by cellular immunity and long-term protection: (1) the functional profile of antigen-specific T cells associated with disease control; (2) if apoptosis is involved in disease severity; (3) if antigen-specific T cell responses have the potential to migrate to the lung and eventually become T$_{RM}$ cells. To this end, we perform detailed phenotypic and functional analyses in clinically defined groups of patients recruited during the first wave of SARS-CoV-2 infection, including the assessment of T$_{RM}$ cells in the lungs of convalescent patients. Informing on the immunological parameters associated with disease control and patient prognosis will aid vaccine development and monitoring of vaccinated individuals towards prediction of immune control.

## Results

**Cohort characteristics.** Patients were recruited during the first pandemic wave of SARS-CoV-2 (spring 2020) at the Hospital Universitari Vall d'Hebron in Barcelona. A total of 46 patients were included, in which 14 individuals were symptomatic non-hospitalized cases, 20 individuals corresponded to mild-hospitalized cases, and 12 to severe-hospitalized cases. Only one patient from the severe group corresponded to a fatal case. Samples were obtained between 7 and 16 days after symptom onset and no differences between groups were detected (Supplementary Fig. 1a). Supplementary table 1 shows a summary of the participant characteristics and baseline determinations, in which significant differences are evidenced between the three groups. As previously reported[26,28,29,35,36], age, lymphopenia, and biochemical parameters such as D-dimer, IL-6, and ferritin were associated with disease severity. Quantification of the viral load between days 5 and 15 after symptom onset is also reported; however, values between the groups were not statistically significant. Some of the clinical parameters used to stratify mild and severe-hospitalized cases are shown in Supplementary Fig. 1b: days to discharge since symptoms onset ($p < 0.0001$, by two-tailed Mann–Whitney U-test), baseline IL-6 ($p = 0.008$, by two-tailed Mann–Whitney U-test) and the percentage of oxyhemoglobin saturation in arterial blood /fraction of inspired oxygen (SAFI) ratio at baseline and after 48 h ($p = 0.0002$, by two-tailed Mann–Whitney U-test). These parameters were used to address associations between immunological parameters and disease severity. In some analyses, 12 control individuals sampled before the COVID-19 pandemic were studied in parallel.

To determine if whole plasma cytokine levels in our groups of COVID-19 patients were similar to previously defined patterns reported before[29,37–40], we analyzed cytokine plasma levels by the ELLA microfluidics platform in the same samples. Levels of IL-1ra, IL-2, IL-6, IL-10, IL-15, chemokine (C-X-C motif) ligand (CXCL)10 (IP-10), IFNγ, granzyme B, and tumor necrosis factor (TNF) were elevated in the plasma of hospitalized groups compared to nonhospitalized patients, with higher levels associated with disease severity (Supplementary Fig. 1c). Of note, the deceased patient from the severe cohort (circled in green) had very high levels of some of the molecules associated with severity and fatality prediction, namely IL-6 and IL-1ra[38,40]. Further, CXCL10 was the most significant predictor of hospitalization during acute infection, as previously observed[13], which may be a reflection of increased IFNγ levels in these patients. C–C chemokine ligand 2 (CCL2), also referred to as monocyte chemoattractant protein 1, was significantly higher in the severe-hospitalized group compared to the nonhospitalized patients, while IL-4, IL-7, IL-13, IL-17A, granulocyte-macrophage colony stimulating factor (GM-CSF) were similar among all three groups of patients (Supplementary Fig. 1c). Strikingly, the levels of IL-12p70 were higher in the plasma of the nonhospitalized compared to the mild COVID-19 group, and the deceased patient had the second lowest level of IL-12p70 of the severe group (0.093 pg/mL; Supplementary Fig. 1c).

**Disease severity and targeted antigen shape T cell function in acute infection.** The functional capabilities of specific CD4+ and CD8+T cells against SARS-CoV-2 were measured by intracellular cytokine staining in samples ranging from 7 to 16 days-post-symptom onset (mean of 12 days) from all three groups. For that, we stimulated peripheral blood mononuclear cells (PBMC) with overlapping membrane (M), nucleocapsid (N), and spike (S)

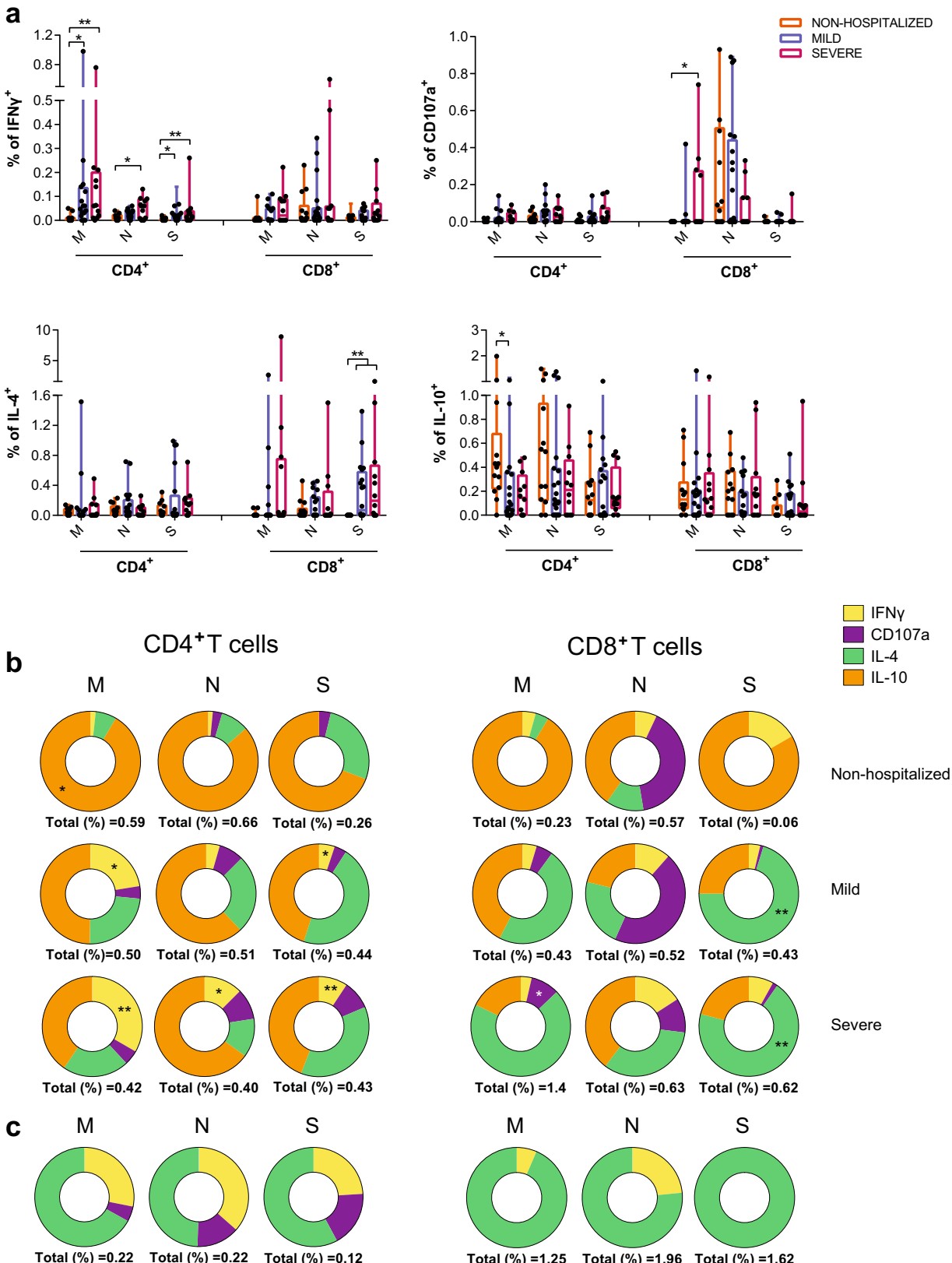

peptide sets and determined the expression of IFNγ, IL-4, and IL-10, along with the degranulation marker CD107a in CD4[+] and CD8[+]T cells (Supplementary Fig. 2a, b). For each function we calculated the net response for each peptide set (background subtracted) and compared these antigen-specific T cell responses among all three groups (Fig. 1a). We also analyzed the overall net

response, which included responses to all peptides, for each function (Supplementary Fig. 2c). This way, differences in the frequency of IFNγ-secreting antigen-specific T cells were significantly higher among the hospitalized groups compared to the outpatients (in CD4[+]T cells: $p = 0.020$ for M and S peptides in the mild group; $p = 0.004$ for M, $p = 0.011$ for N, and $p = 0.007$

**Fig. 1 Functional characteristics of acute SARS-CoV-2-specific T cells. a** Comparison of the net frequency (background subtracted) of interferon (IFN)γ, CD107a, interleukin (IL)-4, and IL-10 expression in SARS-CoV-2-specific CD4+ and CD8+T cells in response to viral proteins (membrane (M), nucleocapsid (N), and spike (S)) between study groups (nonhospitalized $n = 14$ in orange; mild n = 20 in blue and severe $n = 12$ in pink). Statistical comparisons were performed using Kruskal–Wallis rank-sum test with Dunn's multiple comparison test (two-sided): CD4+ IFNγ (M, $p = 0.020$ and $p = 0.004$; N, $p = 0.011$ and S, $p = 0.020$ and $p = 0.007$) and IL-10 (M, $p = 0.035$); CD8+ CD107a (M, $p = 0.037$), IL-4 (S, $p = 0.004$ and $p = 0.003$). **b** Donut charts summarizing the contribution of each function to the overall CD4+- and CD8+-specific T cell response by targeted viral protein and individual group of patients. Data represent the mean value of the net frequency of each function indicated by color code considering all patients, responders and nonresponders. Total response value (%) is shown under each pie chart and represents the cohort average of the overall net frequency considering all individuals and adding up all functions (nonhospitalized $n = 14$; mild $n = 20$ and severe $n = 12$). Statistical comparisons as performed in **a** are indicated with * and ** symbols. **c** Donut charts summarizing the distribution of individual functions among specific CD4+ and CD8+T cells to either the M, N, or S protein from the only fatal case within the severe COVID-19 group.

for S peptides in the severe group; Fig. 1a, by Kruskal–Wallis one-way ANOVA with Dunn's multiple comparisons). Of note, while nonhospitalized patients did not show a significant increase in the production of IFNγ as a group, some individuals did show an increase in their response (>0.02% after background subtraction) representing 43% of responders, which was lower than the 80 and 92% of responders observed in mild and severe-hospitalized groups. In contrast degranulation, measured by CD107a expression, was less detected in general and significance among the groups was only reached in response to M peptides in severe patients compared to nonhospitalized patients (Fig. 1a). We also calculated double positive IFNγ/CD107a CD8+T cells, as a surrogate of cytotoxic polyfunctional cells, since double positive cells could be detected in some patients, as exemplified in Supplementary Fig. 2. Interestingly, while the frequency of double positive CD107a+IFNγ+CD8+T cells responding to the M peptides positively correlated with viral load, the same subset specific for N peptides inversely correlated with baseline levels of IL-6 within the hospitalized cohort (Supplementary Fig. 3a).

Assessment of two other functions, IL-4 and IL-10, demonstrated major dominance of these responses based on the cohort and the viral target. A general induction of an IL-4-specific CD8+T cell response was observed in response to the viral spike in hospitalized patients compared to the nonhospitalized individuals ($p = 0.004$ and $p = 0.003$ for mild and severe patients, respectively, by Kruskal–Wallis one-way ANOVA with Dunn's multiple comparisons; Fig. 1a). Of note, higher levels of spontaneous secretion of IL-4 (in unstimulated conditions) were observed in hospitalized patients, which essentially correlated with the number of days since symptoms onset to discharge and baseline IL-6 levels (Supplementary Fig. 3b). Moreover, SARS-CoV-2 viral load positively correlated with the overall capacity of CD4+T cells to secrete IL-4 in response to TCR independent unspecific activation with PMA/ionomycin (PMA/Io) (Supplementary Fig. 3c). In contrast, the expression of IL-10, a prototypical regulatory cytokine, was significantly increased in CD4+T cells from nonhospitalized patients after stimulation with M peptides when compared to the mild COVID-19 group ($p = 0.035$; Fig. 1a).

Correlations between the net frequency of a given function and clinical parameters were consistent with more CD4+T cells secreting IFNγ and more CD8+T cells secreting IL-4 in response to M and S peptides associated with disease severity (Supplementary Table 2). Even the total CD4+ or CD8+T cell IFNγ response and the total IL-4 secretion by CD8+T cells against any of the three viral proteins (all peptides) correlated with more days at the hospital for IFNγ or with other clinical parameters for IL-4 (Supplementary Table 2). Further, antigen-specific CD4+T cells degranulating in response to all viral peptides and, in the case of CD8+T cells, in response to M peptides also correlated with higher levels of IL-6 (Supplementary Table 2). In contrast, the percentage of M-specific CD4+T cells secreting IL-10 correlated

with better prognosis in all clinical parameters (Supplementary Table 2) and for N-specific positively with better oxygenation at 48 h (Supplementary Table 2).

Actually, when the overall response, including all functions, was represented as donut charts displaying the mean frequency of responses including all individuals (responders and nonresponders), differences among groups in response to each peptide set were emphasized (Fig. 1b). This was, M peptides were shown to mostly stimulate IL-10 secretion in nonhospitalized patients, while in hospitalized cases, increasing amounts of IFNγ for CD4+T cells and of IL-4 and degranulation for CD8+T cell were observed (Fig. 1b). In addition, N peptides induced higher frequencies of antigen-specific CD8+T cells degranulating in mild and nonhospitalized cases, while S peptides stimulated IL-4 secretion mainly in the hospitalized groups (Fig. 1b). Overall, our analyses indicated, on one hand, group-based differences, where a dominance of IL-4 and IFNγ SARS-CoV-2-specific responses were associated with disease severity and of IL-10 to minor disease; on the other hand, we observed targeted protein based differences, where M and N peptides induced a Th1 profile exemplified by IFNγ in CD4+T cells and degranulation (CD107a) in CD8+T cells, respectively, and S peptides induced a biased Th2 profile exemplified by IL-4. This pattern was shown in an exaggerated manner in the deceased patient, in which IL-10 responses were absent and IL-4 together with some IFNγ dominated antigen-specific responses (Fig. 1c).

**Patterns of chemokine receptors associated with SARS-CoV-2-infected patients.** Next, we aimed to determine if part of the specific T cell response was potentially migrating towards the infected tissues by assessing the proportion of CCR7 and CXCR3 expression within the same analyses. In peripheral blood, CCR7 distinguishes T cells homing to lymph node (LN) when expressed, or effector memory (EM) T cell subsets migrating to tissues when absent[41], while CXCR3 may help define antiviral T cells infiltrating inflamed tissues, including the lung parenchyma[25]. CD4+T cells showed only two evident subsets in most patients based on CCR7 expression, since CXCR3 was homogeneously dimly expressed in these two subsets (Supplementary Fig. 2a and 2c) and no differences between the different study groups were observed (Fig. 2a). In contrast, CD8+T cells presented five subsets based on these chemokine receptors (Supplementary Fig. 2a and 2c), and significant differences among the groups were observed (Fig. 2b). Non-hospitalized patients showed increased frequencies of CCR7hCXCR3dCD8+T cells, while severe patients presented increased frequencies of EM CCR7-CXCR3+T cells ($p = 0.0012$ and $p = 0.0034$, respectively, by Kruskal–Wallis one-way ANOVA, Fig. 2b, c). In fact, the accumulation of CCR7hCXCR3dCD8+T cells indicated good prognosis and negatively correlated with the number of days to discharge since symptoms onset and with IL-6 levels at hospital entry (Fig. 2d),

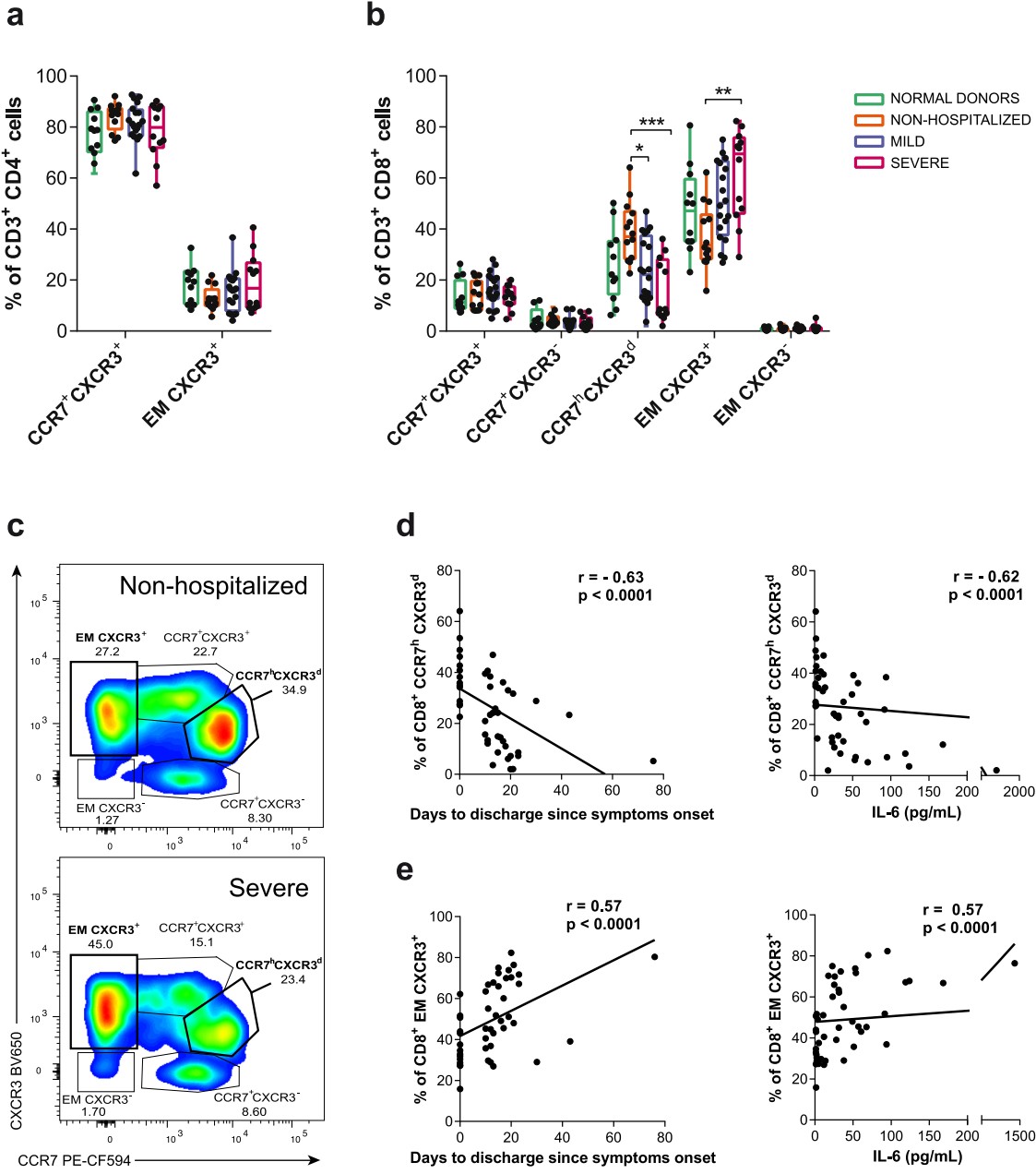

**Fig. 2 T cell migratory patterns during acute SARS-CoV-2 infection. a**, **b** The frequency of various T cell subsets defined by CCR7 and CXCR3 within CD4+ (**a**) and CD8+T cells (**b**). Each dot represents one patient of a specific cohort, indicated by color code (normal donors n = 12; nonhospitalized n = 14; mild n = 20 and severe n = 12). Data are shown as individual patients and boxes and error bars represent median and interquartile range (IQR). Statistical comparisons were performed using Kruskal–Wallis rank-sum test with Dunn's multiple comparison test (two-sided): CCR7hCXCR3d (p = 0.029 and p = 0.0007) and EM CXCR3+ (p = 0.002). **c** Representative flow cytometry plots gating the different CD8+T cell subsets in a nonhospitalized (top) and a severe patient (bottom). **d**, **e** Correlations between days to discharge since symptoms onset or interleukin (IL)-6 baseline levels and the frequency of CD8+ CCR7hCXCR3d (**d**) and CD8+ EM CXCR3+ (**e**) subpopulations. Two-sided spearman rank correlation (n = 46).

while the frequency of EM CXCR3+ CD8+T cells significantly correlated with disease severity parameters (Fig. 2e), results that have also been observed in SARS-CoV-2-specific T cells[42].

We then focused on the distribution of the net antigen-response for each peptide and cohort among these CCR7/CXCR3 subsets. Overall, antigen-specific CD4+T cells showed a distinct pattern based on the function assessed: IFNγ, degranulation (CD107a) and IL-4 were significantly associated to EM CXCR3+ CD4+T cells across the different groups and proteins, while IL-10 was associated to the LN-homing fraction (CCR7+CXCR3+) in response to M and N protein peptides in the

nonhospitalized group (Fig. 3a and Supplementary Fig. 3d). Of note, in general, most individuals in the hospitalized groups also showed this trend for the IL-10 response, although statistical significance was not reached as a group. Moreover, several subsets out of these antigen-specific CD4+T cells, mostly the ones secreting IFNγ or IL-4, correlated with worse prognosis in the clinical parameters assessed before, and some examples are shown in Fig. 3b–3e. In general, stronger associations were observed within the CCR7+CXCR3+ subset, which correlated with severity, except if this subset was secreting IL-10 against M peptides (Fig. 3f). Moreover, SARS-CoV-2 viral load was

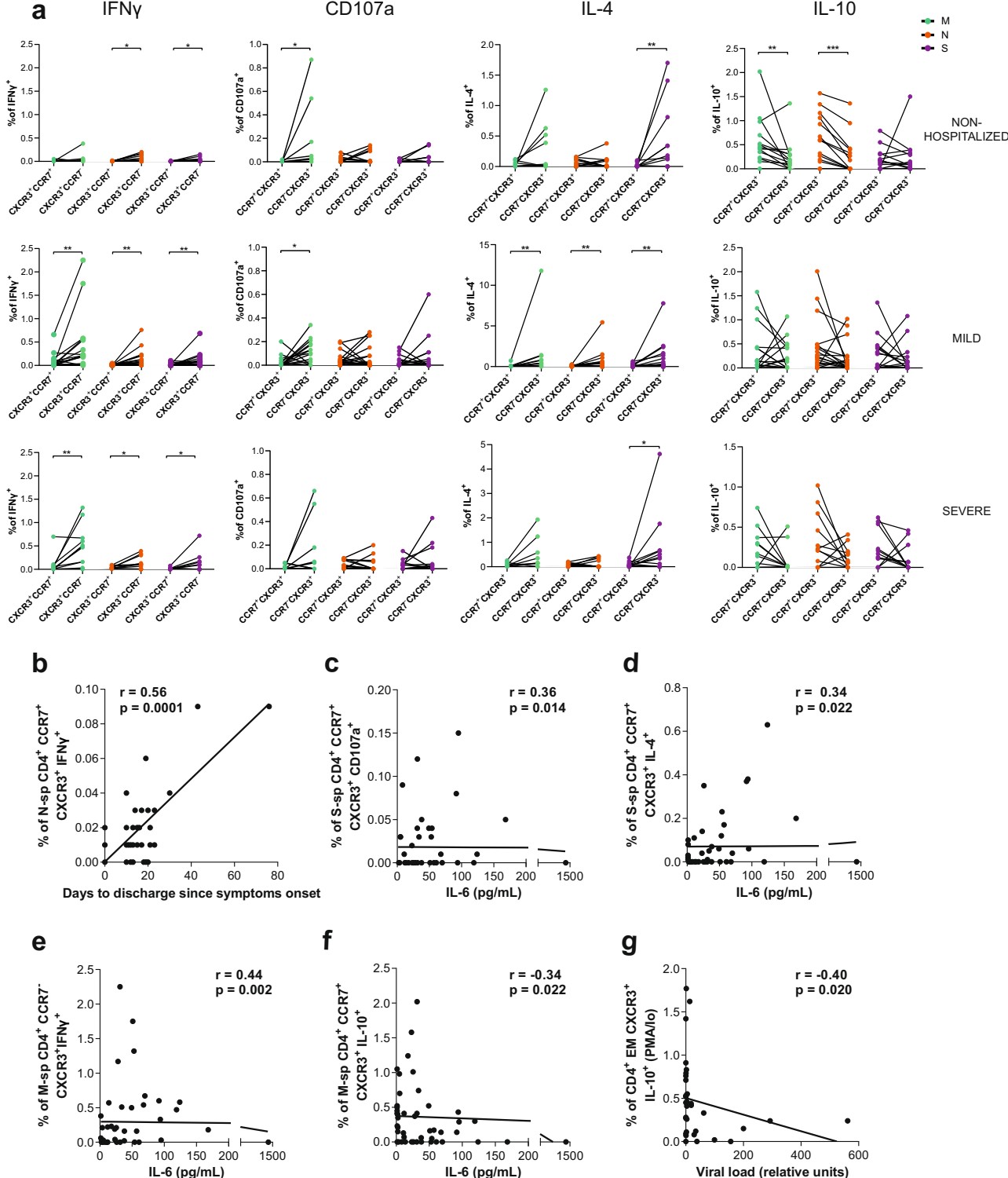

negatively associated with the overall capacity of EM CXCR3+ CD4+T cells to secrete IL-10 in response to TCR independent unspecific activation with PMA/Io (Fig. 3g).

In the analyses of the proportion of antigen-specific CD8+T cells in each CCR7/CXCR3 subset, we did not consider the EM CXCR3- subset, which represented <1% in most patients (Fig. 2b). As expected, IFNγ antigen-specific CD8+T cells were more frequent among the CXCR3+ subsets, with some individual exception, such as N-specific CCR7+CXCR3- T cells within the nonhospitalized group (Supplementary Fig. 4a). Further, IFNγ

secreting CCR7hCXCR3dCD8+T cells in response to N peptides correlated negatively with days to hospital discharge (Supplementary Fig. 4b). Degranulation, which was enhanced also after N stimulation, was unexpectedly detected in all CD8+ CCR7/ CXCR3 subpopulations (Supplementary Fig. 5a). Though only in the LN-homing CCR7+CXCR3+CD8+T cell subset in response to S peptides, degranulation was associated with higher viral load (Supplementary Fig. 5b). With respect to IL-4 secreting antigen-specific CD8+T cells, in general these responses were more frequent in CCR7+ LN-homing subsets, and as mentioned before,

**Fig. 3 Migratory patterns of acute SARS-CoV-2-specific CD4$^+$T cells. a** Net frequency of interferon (IFN)γ, CD107a, interleukin (IL)-4 and IL-10 expression in SARS-CoV-2-specific CD4$^+$ T cells based on CXCR3$^+$CCR7$^+$ and CXCR3$^+$CCR7$^-$ subsets for each individual patient (nonhospitalized $n = 14$; mild $n = 20$ and severe $n = 12$). Viral proteins are shown in color green (membrane protein, M), orange (nucleocapsid protein, N) and purple (spike protein, S). Dots connected by the same line represent the same individual. Statistical comparisons were performed using two-sided nonparametric Wilcoxon matched-pairs signed rank test to compare the two groups (CXCR3 + CCR7$^+$ vs. CXCR3$^+$CCR7$^-$): nonhospitalized (IFNγ, $p = 0.016$ and $p = 0.016$; CD107a, $p = 0.031$; IL-4, $p = 0.008$; IL-10, $p = 0.008$, and $p = 0.0005$), mild (IFNγ, $p = 0.006$, $p = 0.005$, and $p = 0.006$; CD107a, $p = 0.035$; IL-4, $p = 0.004$, $p = 0.008$, and $p = 0.002$), and severe (IFNγ, $p = 0.005$, $p = 0.024$, and $p = 0.039$; IL-4, $p = 0.042$). **b–d** Correlation between the days to discharge since symptoms onset or IL-6 and the frequency of nucleocapsid or spike-specific CD4$^+$ CXCR3$^+$CCR7$^+$ expressing IFNγ (**b**), CD107a (**c**), or IL-4 (**d**). **e**, **f** Correlation between IL-6 and the frequency of membrane-specific CD4$^+$ CXCR3$^+$CCR7$^{-/+}$ expressing IFNγ (**e**) or IL-10 (**f**). **g** Correlation between the viral load and the frequency of CD4$^+$ EM CXCR3$^+$ expressing IL-10 after PMA/Ionomycin stimulation. Two-sided spearman rank correlation ($n = 46$ for all correlations except for viral load (**g**), which corresponds to $n = 33$).

they increased with disease severity (Supplementary Fig. 6a). Consequently, the frequency of IL-4 detected in response to M or S peptides in several of these fractions correlated with disease severity (Supplementary Fig. 6b). Remarkably, SARS-CoV-2-specific CD8$^+$T cells secreting IL-10 were strongly represented among the CCR7$^h$CXCR3$^d$ subset, reaching statistical significance in response to any of the viral proteins within the mild disease cohort and in response to N peptides in nonhospitalized patients, but not in the hospitalized group with severe disease (Fig. 4a). Finally, we detected two additional correlations within the CD8$^+$T cell compartment that were of interest: overall antigen-specific EM CXCR3$^+$ CD8$^+$T cells correlated with higher viral loads if responding to M peptides (Fig. 4b), while the same subset responding to N peptides negatively correlated with IL-6 (Fig. 4c). All together, these results demonstrate individual migratory patterns associated with a given function: whilst most of the functions assessed here were associated with lung homing subsets (CCR7$^-$), IL-10-specific T cells expressed high levels of CCR7. In fact, a strong association towards better disease prognosis was established for an increased proportion of CCR7$^h$CXCR3$^d$ CD8$^+$T cells, which represented a major constant source of IL-10 in CD8$^+$T cells. Further, antigenic stimulation could be driving CCR7$^-$ effector immune responses towards the lung, yet under uncontrolled disease progression, such effector functions seemed to increase in LN-homing CCR7$^+$ subsets.

**Apoptosis is enhanced in T cells during severe infection.** We included caspase-3 in the flow cytometry panel as a surrogate marker of apoptotic cell death activation[43], which was quantified in both antigen-specific and bystander T cells from the different subsets of the study groups (Fig. 1a). Overall expression of caspase-3 in response to stimulation was increased in total CD4$^+$T cells of the severe group after M and PMA/Io stimulation and in CD8$^+$T cells after S stimulation in comparison to the nonhospitalized group (Fig. 5a). Moreover, caspase-3 expression in CD4$^+$ and CD8$^+$T cells after stimulation with S-peptides positively correlated with baseline IL-6 and with the number of hospitalization days for CD8$^+$T cells (Fig. 5b). In addition, the overall frequency of CD4$^+$T cells expressing caspase-3 in response to PMA/Io positively correlated with viral load (Fig. 5b). Further, an increased expression of caspase-3 within the CCR7$^h$CXCR3$^d$ subset was, in general, linked to the disease severity (Fig. 5c), reaching statistical significance after PMA/Io stimulation when comparing the severe and the nonhospitalized groups (Fig. 5c). Consequently, those frequencies in response to stimulation (N, S, or PMA/Io) correlated positively with the number of days at the hospital and with baseline IL-6 (Fig. 5d). Expression of caspase-3 in other CCR7$^+$CD8$^+$T cell subsets in response to N and S peptides also correlated with IL-6 baseline levels in patients (Fig. 5d).

Regarding antigen-specific T cells we detected remarkable differences within the IL-10 secreting T cells. In this sense, increased expression of caspase-3 was distinguished in the hospitalized severe group compared to the nonhospitalized in S-specific IL-10$^+$ CD4$^+$T cells ($p = 0.003$), N-specific IL-10$^+$ CD8$^+$T cells ($p = 0.013$) and even in the overall IL-10 secretion capacity in response to PMA/Io response ($p = 0.036$ for CD4$^+$ and $p = 0.004$ for CD8$^+$, by Kruskal–Wallis one-way ANOVA with Dunn's multiple comparisons; Fig. 5e). Further, significant correlations between apoptosis in IL-10 antigen-specific T cells and several clinical parameters supported these results (Fig. 5f). As for the other functions, we only detected positive correlations for the expression of caspase-3 in baseline and N-specific CD107a$^+$ CD4$^+$T cells with viral load and IL-6 levels, respectively (Fig. 5g). These results indicate increased activation-induced cell death associated with viral replication and disease severity affecting total CD4$^+$ and CD8$^+$T cells, a phenomenon that seems to be modulated by the viral protein targeted. Moreover, CD8$^+$CCR7$^h$CXCR3$^d$ T cells, a major producers of IL-10, appeared to be one of the most affected population.

**Ag-specific T$_{RM}$ responses are present in the lungs of convalescent patients.** In order to demonstrate that antigen-specific T cells detected during acute SARS-CoV-2 infection, not only migrate into the lung parenchyma but also persist as T$_{RM}$, we measured their frequency in lung tissue of seven patients (Supplementary Table 3). These patients, who strongly differed in their SARS-CoV-2 infection profile, successfully recovered and SARS-CoV-2 was not detected in the respiratory tract by RT-PCR before they underwent thoracic surgery for different reasons. Briefly, HL24 patient was a young asymptomatic patient whose blood and lung samples were analyzed 21 days after SARS-CoV-2 laboratory confirmation by RT-PCR. In contrast, samples were analyzed between 6 and 10 months after initial SARS-CoV-2 RT-PCR confirmation for three of the mild cases (HL52, HL65, HL75) and the two severe cases (HL27 and HL69). HL81, another mild case, was analyzed close to 4 months after initial infection. Of note, a more detailed COVID-19 clinical history can be found in the methods section.

Antigen-specific T cell responses were analyzed in total lung CD4$^+$ and CD8$^+$T cells and by three different fractions: CD69$^-$ (non-T$_{RM}$), CD69$^+$ (T$_{RM}$) and a subset within CD69$^+$ cells expressing CD103$^+$ (T$_{RM}$ CD103$^+$) (Supplementary Fig. 7A). Of note, CD69$^+$T cells down-regulated T-bet expression, which has been associated to tissue residency[44]. As shown for one of the convalescent patients with previous severe disease (HL27; Fig. 6a), antigen-specific T cells secreting IFNγ were restricted to the T$_{RM}$ fractions, which in the case of S-specific CD4$^+$T cells represented up to 3.47% of the T$_{RM}$ CD103$^+$ subset. Similarly, in the lung of a mild convalescent patient (HL52; Fig. 6b) N-specific CD8$^+$T cells secreting IFNγ were restricted to the T$_{RM}$ fractions and >40% of

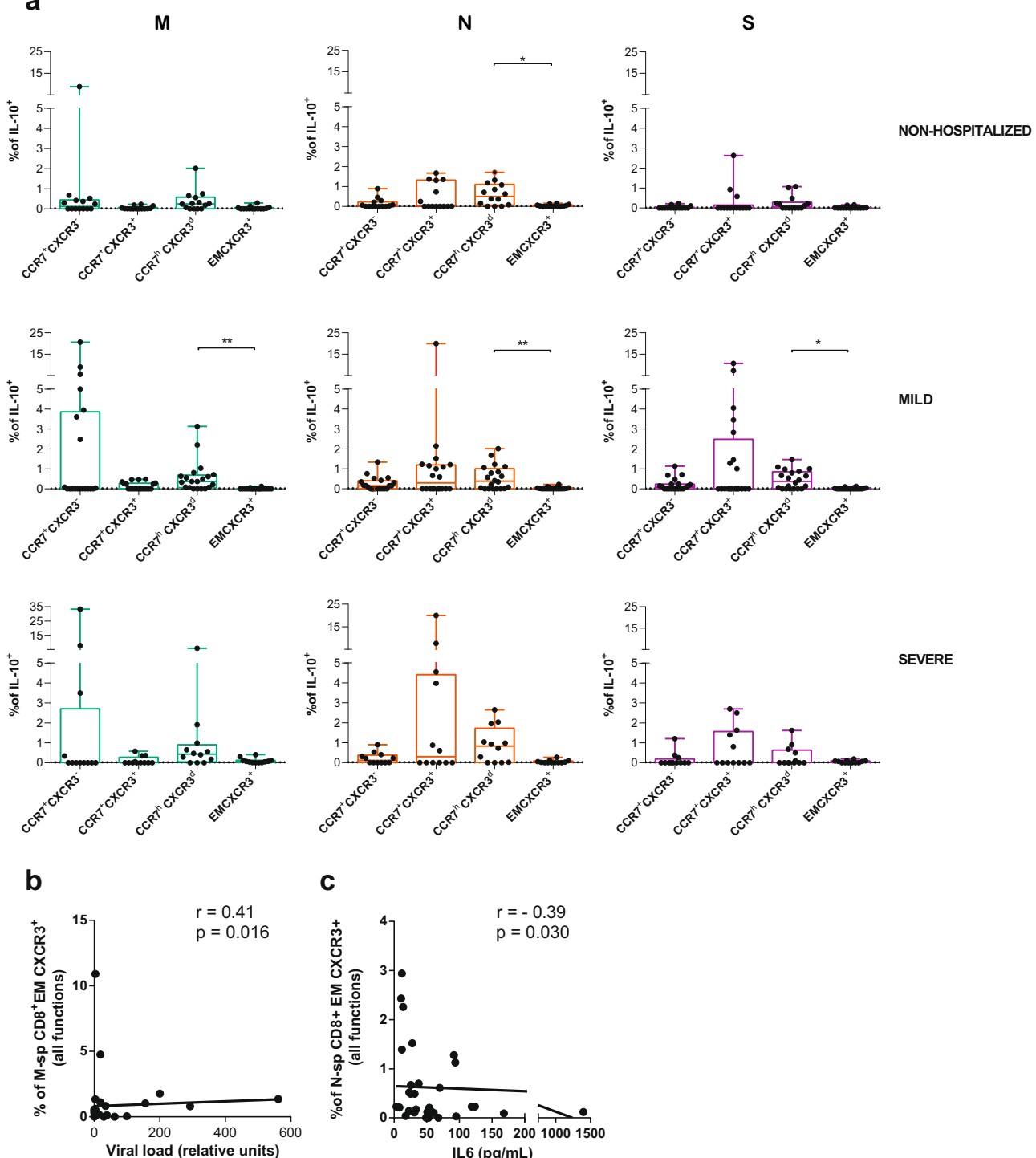

**Fig. 4 IL-10 expression in acute SARS-CoV-2-specific CD8+T cell subsets. a** Net frequency of interleukin (IL)-10 expression in CCR7+CXCR3−, CCR7+ CXCR3+, CCR7hCXCR3d, and EM CXCR3+ subsets within CD8+ T cells after stimulation with any of the three viral proteins (membrane (M), nucleocapsid (N), and spike (S) proteins). Data are shown as median and upper range, where each dot represents an individual patient for each group (nonhospitalized n = 14; mild n = 20, and severe n = 12). Statistical comparisons were performed using Kruskal–Wallis rank-sum test with Dunn's multiple comparison test (two-sided): nonhospitalized (N, p = 0.041) and mild (M, p = 0.003; N, p = 0.007 and S, p = 0.042). **b, c** Correlation between CD8+ EM CXCR3+T cells responding with any function (added net response for interferon (IFN)γ, CD107a, IL-4 and IL-10) against M peptides and viral load (**b**) and against N peptides and baseline IL-6 levels (**c**). Two-tailed spearman rank correlation (n = 33 for viral load and n = 46 for IL-6).

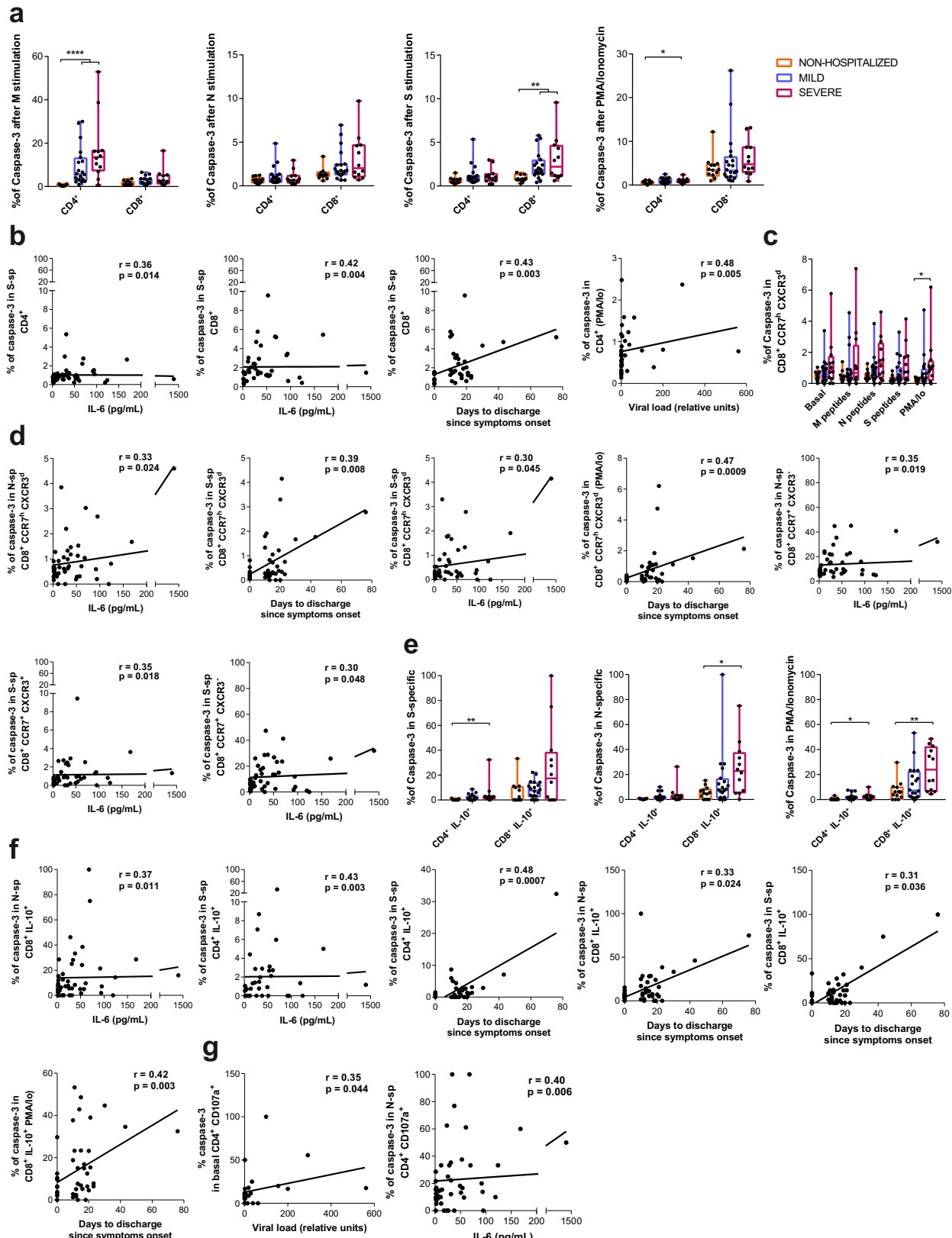

them were simultaneously degranulating (CD107a+). In fact, the assessment of the frequency of cells responding to each peptide pool demonstrated a general trend for more antigen-specific T cells belonging to the $T_{RM}$ subsets for all functions and all patients (Fig. 6c and Supplementary Fig. 7b), which was significant for total CD3+ T cells secreting IFNγ ($p = 0.016$ by

Friedman test; Fig. 6d). Regardless of the variability observed among study patients, the frequency of specific T cell responses increased with disease severity (from left to right; Fig. 6c), while their magnitude was in general low and less consistent for IL-4 or IL-10 responses (Fig. 6c and Supplementary Fig. 7b). Importantly, a consistent polyfunctional IFNγ+CD107a+ T cell response,

**Fig. 5 Caspase-3 expression in T cells during acute SARS-CoV-2 infection. a** Frequency of caspase-3 expression in CD4$^+$ and CD8$^+$T cells after stimulation with membrane (M), nucleocapsid (N) or spike protein (S) and PMA/Ionomycin, in nonhospitalized (orange, $n = 14$), mild (blue, $n = 20$), and severe (pink, $n = 12$) COVID-19 patients. Data are shown as individual patients and boxes and error bars represent median and interquartile range (IQR). Statistical comparisons were performed using Kruskal–Wallis rank-sum test with Dunn's multiple comparison test (two-sided): CD4$^+$ (M, $p < 0.0001$ and $p < 0.0001$; PMA/Io $p = 0.036$) and CD8$^+$ (S, $p = 0.007$ and $p = 0.002$). **b** Correlation between days to hospital discharge since symptoms onset, viral load or baseline interleukin (IL)-6 levels (pg/mL) and the net frequency (background subtracted) of caspase-3 in CD4$^+$ and CD8$^+$T cells after stimulation with the spike protein or PMA/Ionomycin. **c** Frequency of caspase-3 expression in CD8$^+$ CCR7$^h$CXCR3$^d$T cells after stimulation in the same groups as in **a**. Statistical comparisons were performed using Kruskal–Wallis rank-sum test with Dunn's multiple comparison test (two-sided): $p = 0.026$. **d** Correlations between clinical parameters and the net frequency of caspase-3 expression in CD8$^+$ CCR7$^+$ T cell subsets after stimulation. **e** Frequency of caspase-3 expression in IL-10-secreting SARS-CoV-2-specific CD4$^+$ and CD8$^+$T cells responding to the spike protein, the nucleocapsid protein or to PMA/Ionomycin in the same groups as in **a**. Statistical comparisons were performed using Kruskal–Wallis rank-sum test with Dunn's multiple comparison test (two-sided): CD4$^+$ (S, $p = 0.003$; PMA/Io, $p = 0.036$) and CD8$^+$ (N, $p = 0.013$; PMA/Io, $p = 0.004$). **f** Correlation between clinical parameters and IL-10-expressing SARS-CoV-2-specific CD4$^+$ or CD8$^+$T cells, or after PMA/Ionomycin stimulation. **g** Correlation between viral load and the frequency of caspase-3 expression in basal CD107a$^+$ degranulating CD4$^+$T cells and between IL-6 and the net frequency of caspase-3 expression in CD107a-expressing CD4$^+$ in response to N peptides. Two-tailed spearman rank correlation ($n = 46$ for all correlations except for viral load, which corresponds to $n = 33$).

which represented between a median of 0.022 and 0.051% of all CD3$^+$T cells and was mostly associated with the T$_{RM}$ fraction (>75%), was detected against N peptides in all patients except in the asymptomatic (HL24) and the oldest patient (HL75) (Fig. 6e). Of note, the individual patient with a frequent polyfunctional response in blood was HL65, who tested SARS-CoV-2 positive for 4 months,

The comparison between the overall SARS-CoV-2-specific T cell responses detected in lungs with the ones found in contemporaneous peripheral blood samples highlighted strong differences between these two compartments (Fig. 7). For example, in the asymptomatic HL24-patient IFNγ and IL-10 responses were more frequent in blood than in lung, IL-4 was absent, and T cell degranulation was only detectable in lung (Fig. 7). Importantly, this patient was closer to the initial RT-PCR-based laboratory confirmation (3 weeks after), and his profile was more consistent with a nonhospitalized patient. A different pattern was observed for one of the severe convalescent patients (HL27), who had persistent T$_{RM}$ in the lung over-represented by S-specific CD4$^+$T cells secreting IFNγ, which represented up to 1.58% of the total CD4$^+$T cells, while only 0.082% of the circulating CD4$^+$T cells secreted IFNγ and, in this case, in response to M peptides (Fig. 7). In fact, whereas all four functions were detected in T cells from lung after 6 months since initial infection for this patient, they were barely detectable in blood. Last, may be worth mentioning that the highest levels of IL-4 secretion were detected in response to S peptides in the lung of HL65, who tested SARS-CoV-2 positive for 4 months (Fig. 7). Overall, viral-specific T$_{RM}$ responses were detected in all patients. However, no consistent patterns were observed among patients in terms of viral proteins targeted and functions between blood and lung compartments, except for the polyfunctional response detected in tissue against N peptides. Of note, the asymptomatic patient had no detectable antibodies, while all the other patients had detectable antibodies against SARS-CoV-2 in a concomitant plasma sample (measured as total and as IgG fraction). Furthermore, no virus was detected in any of the lung samples by immunofluorescence or viral RNA in situ hybridization. Overall, our results highlight the establishment and persistence of lung-resident T cell immunity against SARS-CoV-2 viral infection. However, in most cases, antigen-specific T$_{RM}$ patterns could not be identified in T cells from blood.

## Discussion

This study identifies unique features of the cellular immunological response against SARS-CoV-2 relevant to infection control and disease progression, which may be critical to informing vaccine assessment and development of new prototypes. First of all, we show that the acute response of nonhospitalized infected patients is characterized by CD4$^+$ and, to less extent, CD8$^+$ SARS-CoV-2-specific T cells secreting IL-10, to which subsets expressing high levels of CCR7 contribute abundantly. In contrast, hospitalized patients show a bias towards an effector response characterized by IFNγ and IL-4 secretion, being the main functions as severity increases. Second, depending on the SARS-CoV-2 viral protein targeted, different CD4$^+$ and CD8$^+$T cell functional profiles are generated, which have clear implications for vaccine design. Third, lymphopenia is partially a consequence of increased apoptosis in antigen-specific and nonspecific T cells, which is associated with disease severity and where SARS-CoV-2-specific subsets, such as IL-10 secreting T cells, appear to be more susceptible. Finally, and most important, SARS-CoV-2 T$_{RM}$ can be established and persist for 10 months after infection; nonetheless, the magnitude and profile of the lung SARS-CoV-2-specific T cells strongly differ from the response detected in blood.

Major efforts have recently centered on the identification and characterization of SARS-CoV-2-specific T cells[4,8–14]. Several of these studies have focused on defining the viral proteins more often targeted by specific T cells, concluding that after infection a broad cell response against multiple structural and non-structural regions of SARS-CoV-2 is detected in most convalescent patients[4,9,12]. More recently, it has been reported that SARS-CoV-2-specific T cells appear to be weaker and less frequent during acute infection[13]. In this sense, in our study, while the frequency of responders based on CD4$^+$T cells specifically secreting IFNγ was similar to previous reports, they were indeed weak in terms of the amount of IFNγ or cytotoxicity. However, SARS-CoV-2-specific T cell response during acute infection appeared to be dominated by IL-4 secretion in hospitalized patients and by IL-10 in nonhospitalized patients. A recent report highlights that, in contrast to other cytokines, increased levels of SARS-CoV-2-specific CD4$^+$T cells secreting IL-10 were mainly detected during active disease[45]. However, these functions have rarely been assessed as part of the specific intracellular T cell response, and others have measured IL-4 in the supernatant of stimulated PBMC without detecting any increase[9,13,14]. In contrast, higher serum levels of IL-4, IFNγ and IL-10 cytokines, among others, have been associated with COVID-19[29,36,37,39]. Differences in methodology and sample timing may account for these discrepancies. In fact, we did not detect changes in plasma levels of IL-4, highlighting differences between measuring the overall level of a given cytokine released systemically rather than the capacity of a small frequency of antigen-specific T cells to quickly secrete such cytokine. Importantly, several studies have detected a negative impact of IL-4 mediated responses on

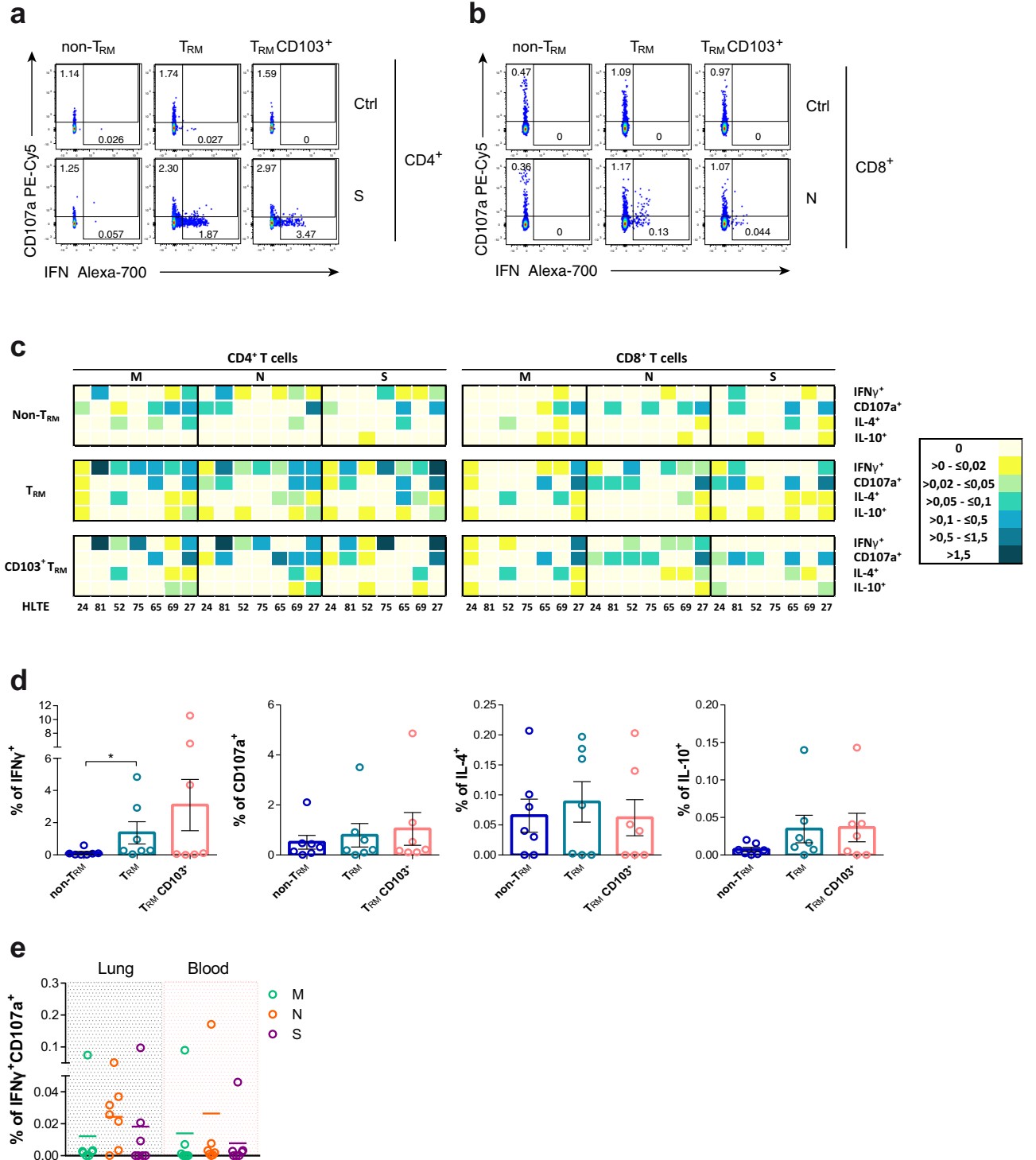

**Fig. 6 Functional analysis of lung-resident SARS-CoV-2-specific T cells. a, b** Flow cytometry plots showing the frequency of interferon (IFN)γ and degranulation (CD107a) by non-resident memory T cells (T_RM), T_RM or CD103+ T_RM in CD4+ from HL27 after spike stimulation and control (**a**) and in CD8+ from HL52 after nucleocapsid stimulation and control (**b**). **c** Heatmaps summarizing the net frequencies of IFNγ, CD107a, interleukin (IL)-4, and IL-10 SARS-CoV-2-specific CD4+ or CD8+ non-T_RM, T_RM, and T_RM CD103+ from seven different SARS-CoV-2 recovered patients. Cytokine production or degranulation are displayed as colors ranging from yellow to blue, based on the frequency, as shown in the key. **d** Net frequency of SARS-CoV-2-specific CD3+T cells in response to all viral proteins (membrane (M), nucleocapsid (N) and spike (S)) from from lung by non-T_RM, T_RM or CD103+ T_RM are shown as mean ± SEM (n = 7). Statistical comparisons were performed using Kruskal–Wallis rank-sum test with Dunn's multiple comparison test (two-sided): p = 0.023. **e** Net frequency of double positive IFNγ/CD107a CD3+T cells from lung or blood after stimulation with membrane (M; green), nucleocapsid (N; orange), or spike protein (S; purple) (n = 7).

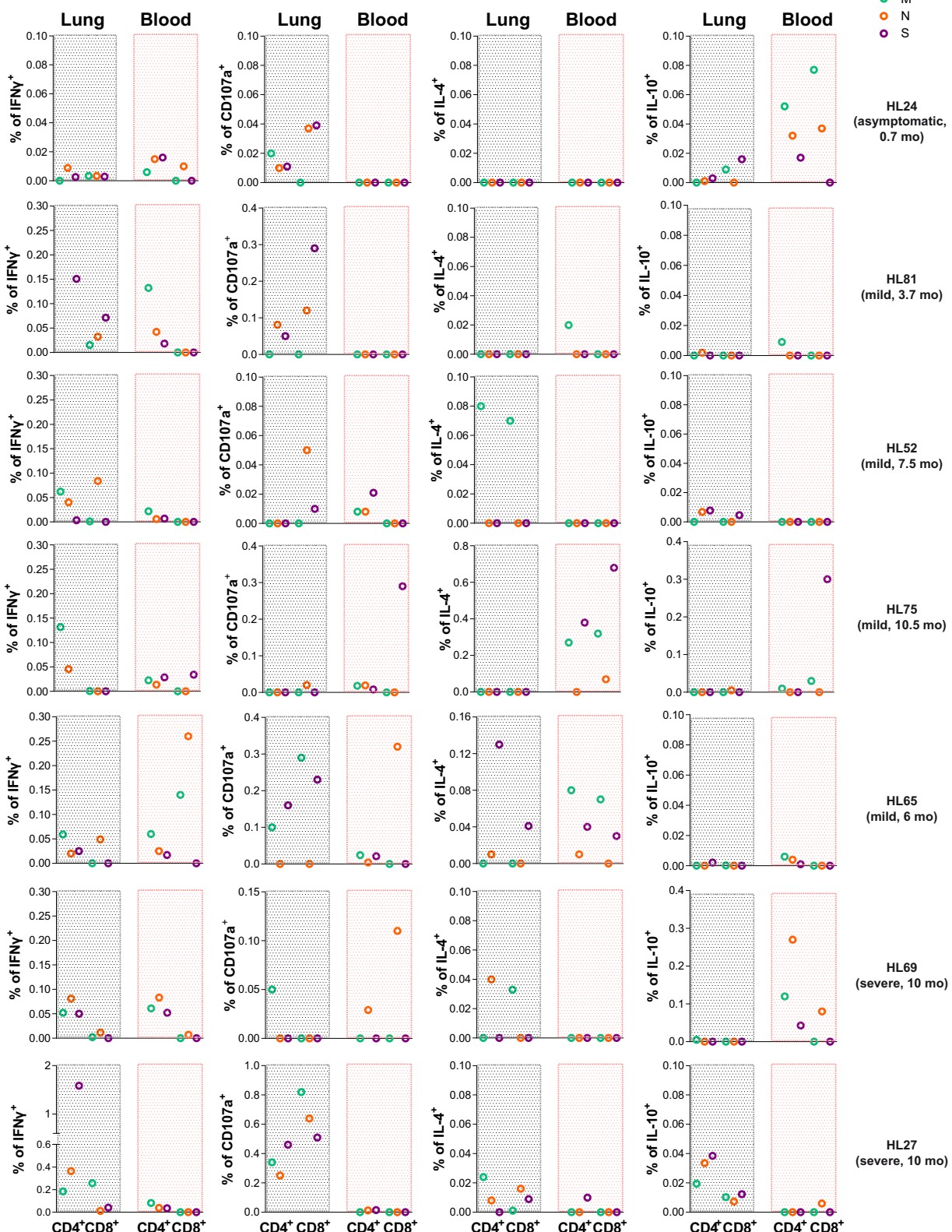

**Fig. 7 SARS-CoV-2-specific T cells in lungs and blood of convalescent patients.** Total CD4$^+$ and CD8$^+$T cell net frequencies of interferon (IFN)γ, CD107a, interleukin (IL)-4, and IL-10 expression in SARS-CoV-2-specific T cells derived from lung or blood from the same patient. Each patient ($n = 7$) is labeled on the right and includes disease profile and sampling time: months (mo) after initial SARS-CoV-2 detection by PCR. Viral proteins are shown in color green (membrane protein, M), orange (nucleocapsid protein, N), and purple (spike protein, S).

immune protection[46]. Furthermore, an increased expression of IL-4 was detected in the lungs of patients who died from SARS-CoV-2 infection[47]. In our cohort, spontaneous IL-4 secretion from both T cell subsets correlated with disease severity, and responses against the spike protein strongly stimulated this response, potentially suggesting the induction of a stronger antibody-directed response. Last, the patient with mild infection who tested positive for 4 months (HL65) showed the most frequent IL-4 response in the lung, which was against S peptides.

Caspase-mediated apoptosis in the immune system is a major contributor to immune homeostasis in a process termed activation-induced cell death[48], which could potentially contribute to minimize an overwhelming cytokine response. Our results are consistent with apoptosis significantly contributing to the lymphopenia detected in COVID-19 patients, and the preferential loss of CD8+T cells is accompanied by an increase in caspase-3 within this compartment, consistent with a pre-publication[49]. Since inflammatory molecules can often be potent activators of cell death, increasing levels of IL-10 may moderate the extent of apoptosis induced, as occurs in mouse models of bacterial infection[50]. Moreover, not only may the inflammatory environment contribute to a higher proportion of bystander T cells succumbing to apoptotic cell death, but viral proteins may also induce apoptosis in both antigen and non-antigen-specific T cells. While effector T cells with degranulation capacity are expected to be more terminally differentiated and, consequently, may be more prone to activation-induced cell death, the fact that other subsets (i.e., CCR7hCXCR3d/IL-10 secreting) were more affected is intriguing and requires further study.

Critically, the only cytokine that was higher in nonhospitalized compared to hospitalized groups was IL-12p70 (Supplementary Fig. 1c), suggesting that, as occurs with other respiratory viruses[51], cell-mediated Th1 immunity is necessary for recovery from respiratory infection. A general predominance of a Th1 profile does not compromise the generation of neutralizing antibodies[46]. Further, it is increasingly accepted that Th-cell subsets are plastic, especially during responses to pathogens in vivo, and even cytokines such as IL-10 can be produced by subpopulations of cells within multiple effector subsets[46]. However, SARS-CoV-2-specific T cells from acute responders demonstrated a biased Th1 profile, where IFNγ-secreting T cells could also secrete IL-2 in the case of CD4+, and TNF and granzyme B (but not IL-10) in the case of CD8+[13]. Our data also show marginal co-expression between IFNγ, IL-4, and IL-10, suggesting multiple polarizations of antigen-responding T cells[46], which is also supported by different CCR7/CXCR3 subsets being the main contributors to a given response. Moreover, this polarization was partially induced by the targeted protein, where M peptides induced the strongest IFNγ secretion in CD4+T cells, N peptides enhanced cytotoxicity in CD8+T cells and S peptides had an overall predominant Th2 profile (IL-4). Similar to what was observed for convalescent patients[9], M and N-specific responses dominated in nonhospitalized and mild-hospitalized cases, indicating that a candidate vaccine including only SARS-CoV-2 spike would limit the array of responses during natural infection[9]. Furthermore, a comparison between the functional profile of antigen-specific CD4+ and CD8+T cells based on the viral protein targeted evidenced a wider functional profile for CD8+T cells targeting M or N peptides compared to S, which was more often observed in milder cases than in severe ones[12] and agrees with recent identification of responses to frequently recognized CD8+T cell epitopes[52]. Overall, our results concur with a broader Th profile in CD8+T cells induced by the non-spike viral proteins and associated with a less severe infection, while spike responses dominated by IL-4+ CD8+T cells

represented a hallmark of disease severity, being the sole response in the fatal case. Further, the observation that N-specific EM CXCR3+CD8+T cell responses with an IFNγ+CD107a+ Th1 profile were associated with nonhospitalization during symptomatic COVID-19 suggests a favorable immune response when the N protein is targeted by CD8+T cells, and that these responses can migrate towards infected tissues. The CXCR3-CXCL10 axis appears critical for the recruitment of CD4+ and CD8+T cells that control influenza and tuberculosis infection in the lung, respectively[25,53]. Consequently, lung T cell recruitment may partially contribute to the lymphopenia detected in patients[54], where this antiviral response will likely establish as resident memory cells. In fact, N-specific IFNγ+CD107+ T_RM were detected in five convalescent patients several months after infection.

Recent deep immune profiles of COVID-19 patients have identified T-bet expression as a transcriptional factor associated to patients with better prognosis[55]. Importantly, T-bet is not only a key regulator of Th1 immune effector responses and CXCR3 expression, essential for effective clearance of pathogens and maintenance of immunity[56], but also crucial for migration, proliferation and survival of T regulatory (Treg) cells during Th1-mediated immune responses in vivo[57]. Indeed, CXCR3 is also found on a subset of CD4+Foxp3+T cells, and the control of inflammatory responses at mucosal surfaces requires IL-10 producing Treg cells[57]. One of the main correlates of disease control during acute infection was IL-10 secretion, which dominated the specific immune response of nonhospitalized patients. Two previous studies have detected reduced frequencies of Treg cells in severe COVID-19 cases[26,58]; however, IL-10 was detected in supernatants from stimulated PBMC of severe patients during acute infection[14], and a similar trend was observed in a cohort of acute patients with a wider range of sampling (4–37 days)[13]. While, as detected in the plasma of our own cohort of patients, increased serum levels of IL-10 have been widely associated with COVID-19 and disease severity (reviewed in[37]), several factors may explain these results. The plasma source of IL-10 can have multiple cellular origins, ranging from myeloid subsets to epithelial cells, since not only may almost all leukocytes produce IL-10, but also the range amplifies during inflammation[50]. The systemic increase in IL-10 may still act as a compensatory response to limit massive ongoing inflammation in severe patients[59]. We propose that an early effector-specific T cell response coordinated with engagement of other immune profiles limiting inflammation may aid at promoting infection resolution. This notion is supported by comprehensive analyses of common immune correlates of protection from mortality in mouse models of influenza and SARS-CoV infection, which revealed a unique T regulatory suppressive profile that contributed to this balance[60]. Furthermore, it has previously been shown that Treg activity is required during viral infections to allow for appropriate generation and migration of immune effector cells to the site of infection[61,62], while blocking the action of the IL-10 secreted by antiviral T cells results in enhanced pulmonary inflammation and lethal injury[17,50,63]. In fact, during influenza infections, type I IFN signaling may contribute to IL-10-producing lymphocyte recruitment to the site of infection to moderate excessive inflammation, which will be coincident with the onset of the adaptive immune response[50], being CD8+T cells a primary source of IL-10 production in the respiratory tract[64]. Accordingly, while IL-10 plasma levels increased with disease severity, the fatal case had one of the lowest levels in plasma, which was accompanied by an absolute lack of IL-10 secretion by antigen-specific T cells.

T_RM strategically residing in peripheral tissues are key to controlling mucosal infections and providing rapid and durable

immunity against reinfection[20,65]. Previous studies in SARS-recovered patients already pointed towards persistence of a memory T cell response for up to 6 years after infection, and suggested vaccine-mediated induction of $T_{RM}$ as a long-term protection strategy[5]. In concordance, a larger proportion of CD8$^+$T cell effectors with $T_{RM}$ characteristics were present in bronchoalveolar lavages from patients with moderate infection compared to severe-infected patients[66]. We indeed report the existence of a high frequency of SARS-CoV-2-specific $T_{RM}$ in the lung of some patients infected 6–10 months before. In particular, for the severe and durable COVID-19 patient (HL27), while all functions were represented in $T_{RM}$ in a remarkably higher proportion than in blood, IFNγ in response to S peptides dominated. In this sense, high frequency of spike protein-specific CD4$^+$T cell responses was observed in blood in patients who had recovered from COVID-19[4,9,12]. Importantly, CD4$^+$T cells are necessary for the formation of protective CD8$^+$T$_{RM}$ during influenza infection, and cytokines, such as IFNγ, are necessary signals for this process[20]. However, CD4$^+$T cells themselves can be cytotoxic and, actually, have been shown to confer protection against influenza[20]. We also detected degranulation in response to viral peptides in CD4$^+$ and even more so in CD8$^+$T cells from the lung, which in the case of the asymptomatic young patient were completely absent in blood. Lack of degranulation in blood from convalescent patients has also been reported[12]. Further, the fact that the lung sample of this young asymptomatic patient was 3 weeks after RT-PCR laboratory confirmation of infection, suggests early recruitment of cytotoxic T cells to the lung even in asymptomatic cases. Moreover, the most frequent responses by circulating T cells from this asymptomatic patient were CD4$^+$ and CD8$^+$T cells secreting IL-10, which concurs with the dominating pattern in nonhospitalized patients during acute infection.

We acknowledge that our study has several limitations, one being that sample size for the different groups was small to be conclusive. However, this was compensated by a narrow window of sampling during acute infection (7–16 days, post-symptoms onset) and by a comprehensive clinical characterization to stratify patients to study groups. In this sense, multiple correlations support our main findings and provide strength to our data, which is also largely supported by current literature. Further, identification of the precise phenotypes generating antigen-specific T responses, such as Treg cells for IL-10, or the consideration of other T lymphocytes such as γδ T cells should also be considered in future studies. Last, only seven lung samples obtained from different COVID-19 convalescent patients could be studied. While these patients are so far scarce, the immune responses identified in those samples not only contributed to round out the present report, but also represent the first evidence to our knowledge of persisting SARS-CoV-2-specific $T_{RM}$ in the lung. Disease severity during acute SARS-CoV-2 infection is associated with strong peripheral T and B cell responses[4,27], which not only may relate to antigenic burden but, also could potentially translate into a significant proportion of antigen-specific $T_{RM}$ in the lung once the patient recovers. Remaining important questions concern the level of viral replication and associated symptomatology that will stimulate an effective immune response at the respiratory tract, and also, how quick this response will be established. However, the fact that an asymptomatic patient had degranulating antigen-specific T cells in the lung 3 weeks after infection is at least encouraging. Overall, a balanced effector/anti-inflammatory response may be key for early viral containment, where antigen-specific IL-10$^+$ T cells could be determinant in limiting inflammation. Thus, the possibility that overstimulated proinflammatory T cells contribute to disease severity cannot be ruled out. Our findings encourage next-generation vaccine designs to consider including viral proteins beyond the spike protein, in particular nucleocapsid peptides, which should broaden and balance the functional profile of memory T cells, resembling control of natural infection.

## Methods

**Ethics statement.** This study was performed in accordance with the Declaration of Helsinki and approved by the corresponding Institutional Review Board (PR(AG) 192/2020, PR(AG)212/2020, PR(AG)116/2018, and PR(AG)117-2018) of the Vall d'Hebron University Hospital (HUVH), Barcelona, Spain. Written informed consent was provided by all patients recruited to this study and samples were prospectively collected and cryopreserved in the Vall d'Hebron Research Institute.

**Healthy donors.** Blood samples from healthy adult donors were obtained via phlebotomy. These blood samples were collected for studies unrelated to COVID-19 between September 2018 and June 2019. At the time of enrollment in the initial studies, all individual donors provided informed consent that their samples could be used for future studies. These samples were considered to be from unexposed controls given that SARS-CoV-2 emerged as a novel pathogen in December 2019 and these samples were largely collected before this date. These donors were considered healthy in that they had no known history of any significant systemic illnesses. The cohort of healthy donors includes 12 individuals.

**Patients with SARS-CoV-2 infection.** Adult patients, 18 years old and older, diagnosed with acute COVID-19 were recruited at the Vall d'Hebron Hospital during the first COVID-19 outbreak between March and May 2020. Diagnosis of acute COVID-19 was defined by symptomatology and/or clinical findings and confirmed by positive reverse-transcriptase polymerase chain reaction (RT-PCR) for SARS-CoV-2 in a respiratory tract specimen. Immunocompromised patients in which the immune response may be affected were excluded of the study. Twenty milliliters of blood were collected at baseline by phlebotomy in two EDTA tubes and stored at room temperature briefly prior to processing for PBMC and plasma isolation. Samples were obtained between 7 and 16 days after symptoms onset. Biochemistry analyses were measured at baseline in all patients and, subsequently, according to the clinical care needs of each patient during infection. Routine clinical laboratory analyses included complete blood count, coagulation testing (including D-dimer measurement), liver and renal function, electrolytes and inflammatory profile (including C-reactive protein, fibrinogen, ferritin, and IL-6). The study cohort consisted of 12 patients with severe disease, 20 with mild disease, and 14 nonhospitalized individuals. Patient information is summarized in Supplementary table 1. According to disease severity patients, at the discretion of the treating physician, patients were classified in three groups:

(a) Patients with severe disease: individuals with radiologically confirmed pneumonia that required hospitalization and had acute respiratory failure and/or analytical parameters of severity and/or extensive radiological involvement.

(b) Patients with mild disease: individuals with radiologically confirmed pneumonia that required hospital admission but without criteria of severity.

(c) Nonhospitalized patients: individuals without pneumonia and with pauci-asymptomatic disease that did not require hospitalization and managed on an outpatient clinic.

Data were collected prospectively from the medical charts of the patients. We collected sociodemographic characteristics, past medical records, Charlson comorbidity score, concomitant medication, treatments against SARS-CoV-2 infection, adverse drug events, blood test results, imaging studies, microbiological tests, and supportive measures needed. Vital signs, symptoms and physical examination were recorded. Laboratory, microbiology, and imaging studies were performed according to the clinical care needs of each patient.

**Lung samples.** Lung tissue was obtained from seven patients recovered from SARS-CoV-2 infection who needed a lung resection (Supplementary table 3). Analyses were performed using healthy areas from the lung resection. HL24 sample corresponded to a 21-year-old smoking man who had detectable viral load by RT-PCR without symptoms, followed by two negative RT-PCR measurements (8 and 18 days after the positive result). He underwent surgery for pneumothorax 21 days after the positive SARS-CoV-2 detection. HL52, HL75, and HL81 corresponded to a 71, 77, and a 64-year-old men hospitalized for 5, 13, and 9 days, respectively, for mild disease (baseline IL-6 of 26.15, 9.13, and 309 pg/mL, respectively). Both patients underwent thoracic surgery due to lung carcinoma 7.5, 10.5, and 3.7 months after confirmatory RT-PCR. HL65 was a 52-year-old woman hospitalized for three days (without oxygen requirements), with confirmatory SARS-CoV-2 RT-PCR and a baseline IL-6 of 5.25 pg/mL. During the next 4 months, she tested positive for RT-PCR. She underwent thoracic surgery for a pulmonary nodule 2 months after testing negative for RT-PCR (and 6 months after initial discharge). HL69 was a 69-year-old man hospitalized for severe COVID-19 for 35 days (baseline IL-6 of 34.66 pg/mL) and treated for the symptomatology derived of the SARS-CoV-2 infection (with confirmatory RT-PCR). During a post-COVID

examination, a pulmonary nodule was diagnosed, which instigated thoracic surgery 10 months after initial infection. HL27 was a 68-year-old ex-smoker man hospitalized for one month due to respiratory insufficiency caused by SARS-CoV-2 infection (with confirmatory RT-PCR and a baseline IL-6 of 133.7 pg/mL). One month after discharge, he presented a rash in photoexposed skin, potentially related to persistent SARS-CoV-2 infection[67] or treatment with hydroxychloroquine[68]. RT-PCR for SARS-CoV-2 tested positive again, turning into a RT-PCR negative 2 weeks after. During this time, a lung carcinoma was diagnosed, which instigated thoracic surgery 3 months after the negative RT-PCR (and 5 months after initial discharge).

Concomitant to the lung samples, blood samples were also collected. Thus analyzed paired blood and tissue samples corresponded to 21 days after the first positive SARS-CoV-2 detection for the HL24 patient (asymptomatic), 3.7 months after initial infection for HL81 (mild), and between 6 and 10 months after initial infection for HL65 and HL75 (mild), HL52 (mild-PCR+ for 4 months), HL69 (severe), and HL27 (severe-PCR+ for 2 months) patients. Blood samples were immediately processed for PBMC and serum isolation, which was used for SARS-CoV-2 antibody detection.

**SARS-CoV-2 RT-qPCR.** Upper (naso/oropharyngeal swabs) and lower (bronchoalveolar lavage, tracheal aspirate, sputum, or bronchoaspirate) respiratory tract specimens from subjects with suspicion of COVID-19 were received and tested at the Respiratory Viruses Unit of the Microbiology Department of the HUVH. COVID-19 diagnosis was performed by two commercial RT-PCR-based assays, Allplex™ 2019-nCoV (Seegene, Korea, Cat. #RP10244Y) or Cobas® SARS-CoV-2 (Roche Diagnostics, USA, Cat. #09175431190) tests. In addition, an in-house PCR assay using the primer/probe set targeting the nucleocapsid protein (N1) and the human RNase P (housekeeping gene), from the CDC 2019-nCoV Real-Time RT-PCR Diagnostic Panel (Qiagen, Hilden, Germany), was performed. Details on the primers and probes used are shown in Supplementary Table 4. In order to minimize variations due to a nonstandardized collection of a heterogeneous specimen, the Ct values of the viral target were normalized to the housekeeping gene based on the $2^{-\Delta Ct}$ method, where $\Delta Ct$ corresponds to the formula Ct sample − Ct housekeeping.

**Plasma cytokine quantification.** Plasma obtained from our three cohorts of COVID-19 patients ($n = 46$) was analyzed using Ella® platform (Bio-Techne, Minneapolis, Minnesota, USA) for the quantification of the following cytokines and chemokines: CCL2, GM-CSF, IL-10, IL-12p70, IL-1ra, IL-6, IL-7, TNF, CXCL10, Granzyme B, IFNγ, IL-13, IL-15, IL-17A, and IL-4. Samples were 1:2 diluted with sample diluent provided by the manufacturer and loaded onto multiplex cartridges according to manufacturer's instructions prior to their analysis. Results are expressed as pg/mL.

**Phenotyping and intracellular cytokine staining in blood.** PBMCs were isolated from blood by density-gradient centrifugation using Ficoll-Paque and immediately cryopreserved and stored in liquid nitrogen until use in the assays. Cells were thawed the day before the assay and cultured in a T-25 flask at 37 °C with RPMI 1640 (Gibco) supplemented with 10% Fetal Bovine Serum (FBS) (Gibco), 100 μg/ml streptomycin (Fisher Scientific), and 100 U/ml penicillin (Fisher Scientific) (R10). Next day, previous to SARS-CoV-2-peptide pool stimulation, cells were stained with CCR7 (PE-CF594, BD Biosciences) and CXCR3 (BV650, BD Biosciences) for 30 min at 37 °C. After a washing with PBS, PBMCs were stimulated in a round-bottom 96-well plate for 5 h at 37 °C with 1 μg/ml of SARS-CoV-2 peptides (PepTivator SARS-CoV-2 M, N and S, Miltenyi Biotec) in the presence of 1 μl/ml of Brefeldin A (BD Biosciences), 0.7 μl/ml of Monensin (BD Biosciences) and 3 μl/ml of α-CD28/CD49d (clones L293 and L25, BD Biosciences). Anti-CD107a (PE-Cy5, BD) was also added at this time. For each patient, a negative control, cells treated with medium, and positive control, cells incubated in the presence of 81 nM PMA and 1 μM Ionomycin, were included. After stimulation, cells were washed twice with PBS and stained with Aqua LIVE/DEAD fixable dead cell stain kit (Invitrogen, Cat. #L34966). Cell surface antibody staining included anti-CD3 (PerCP), anti-CD4 (BV605), and anti-CD56 (FITC) (all from BD Biosciences). Cells were subsequently fixed and permeabilized using the Cytofix/Cytoperm kit (BD Biosciences, Cat. #554714) and stained with anti-Caspase-3 (AF647, BD Biosciences), anti-Bcl-2 (BV421, Biolegend), anti-IL-4 (PE-Cy7, eBioscience), anti-IL-10 (PE, BD Biosciences), and anti-IFNγ (AF700, Invitrogen) for 30 min. Cells were then fixed with PBS 2% PFA and acquired in a BD LSR Fortessa flow cytometer (Cytomics Platform, High Technology Unit, Vall d'Hebron Institut de Recerca). FACS Diva software 6.2. (firmware version 1.2) was used for flow cytometry sample acquisition and analyzed with FlowJo v10.7.1 software (TreeStar). FMO controls were used to draw the gates for each function.

For the patients with lung samples, their contemporary blood sample was processed immediately and the T cell response assay was performed without a previous cryopreservation step. Isolated PBMC were rested for 4 h in the incubator and then stimulated with the same SARS-CoV-2 peptides (M, N, and S) overnight, following the same protocol described above with half the amount of Brefeldin A and Monensin to avoid toxicity.

**Phenotyping and intracellular cytokine staining in lung.** Lung tissues were collected in antibiotic-containing RPMI 1640 medium from the Thoracic Surgery Service at the Vall d'Hebron University Hospital. Immediately following surgery, the tissue was dissected into approximately 8-mm³ blocks. These blocks were first enzymatically digested with 5 mg/ml collagenase IV (Gibco) and 100 μg/ml of DNase I (Roche) for 30 min at 37 °C and 400 rpm and, then, mechanically digested with a pestle. The resulting cellular suspension was filtered through a 70 μm pore size cell strainer (Labclinics), washed twice with PBS and cultivated with R10 in a round-bottom 96-well plate overnight at 37 °C with 1 μg/ml of SARS-CoV-2 peptides (M, N, and S) in the presence of 3 μL/mL α-CD28/CD49d (clones L293 and L25, BD Biosciences), 0.5 μL/mL Brefeldin A (BD Biosciences), 0.35 μL/mL Monensin (BD Biosciences) and 5 μL/mL anti-CD107a-PE-Cy5. For each patient, a negative control, cells treated with medium, and positive control, cells incubated in the presence of 40.5 nM PMA and 0.5 μM Ionomycin, were included. Next day, cellular suspensions were stained with Live/Dead Aqua (Invitrogen) and anti-CD103 (FITC, Biolegend), anti-CD69 (PE-CF594, BD Biosciences), anti-CD40 (APC-Cy7, Biolegend), anti-CD8 (APC, BD Biosciences), anti-CD3 (BV650, BD Biosciences), and anti-CD45 (BV605, BD Biosciences) antibodies. Cells were subsequently fixed and permeabilized using the Foxp3 Staining Buffer Set (Thermo Fisher Scientific, Cat. #00-5523-00) and stained with anti-IL-4 (PE-Cy7, eBioscience), anti-IL-10 (PE, BD Biosciences), anti-T-bet (BV421, Biolegend), and anti-IFNγ (AF700, Invitrogen) antibodies. After fixation with PBS 2% PFA, cells were acquired in a BD LSR Fortessa flow cytometer. FACS Diva software 6.2. (firmware version 1.2) was used for flow cytometry sample acquisition and analyzed with FlowJo v10.7.1 software (TreeStar).

**SARS-CoV-2 serology.** Serological status of HL24, HL27, HL52, HL65, HL69, HL75 and HL81 patients was determined in serum using two commercial chemiluminescence immunoassays (CLIA) targeting specific SARS-CoV-2 antibodies: (1) Elecsys Anti-SARS-CoV-2 (Roche Diagnostics, USA, Cat. #09203079190) was performed on the Cobas 8800 system (Roche Diagnostics, USA) for qualitative determination of total antibodies (including IgG, IgM, and IgA) against nucleocapsid SARS-CoV-2 protein; and (2) Liaison SARS-CoV-2 S1/S2 IgG (DiaSorin, Italy, Cat. #311510) was performed on the LIAISON XL Analyzer (DiaSorin, Italy) for quantitative determination of IgG against the spike (S) glycoprotein subunits 1 and 2 (S1/S2).

**SARS-CoV-2 detection by immunofluorescence and RNA hybridization.** Paraffin-embedded lung tissue samples were processed and analyzed at the Pathology Department of the HUVH. For SARS-CoV-2, lung tissue sections of 3 μm were deparaffinized with xylene and dehydrated in ethanol. Samples were pretreated with CC2 (pH = 6), and rinsed with working PBS. SARS-CoV-2 (SARS-CoV Nucleoprotein/NP Antibody, Rabbit PAb, 6F10, Sino Biological, dilution at 1:1000) antibody was applied and incubated during 1 h. After washing in PBS, slides were mounted in 80% glycerol and sealed. Images were taken using BenchMArk Ultra Ventana System.

RNA hybridization was performed using RNAscope VS Universal Assays and the Ventana Discovery Ultra System. A high sensitivity target-specific probes to SARS-CoV-2 mRNA sequence (probe V-nCoV2019-S, ADC Biotecnec biology) were used. Lung tissue sections of 3 μm tissue sections were mounted on Superfrost Plus microscope slides (Fisher Scientific). The assay was performed according to manufacturer's instructions. Briefly, samples were deparaffinized and pretreated as mentioned. Next, probes were incubated for 2 h at 40 °C and samples were stored overnight in 5× saline sodium citrate buffer. The following day, amplification and signal development was performed by sequential incubation of Pre-Amplifiers, Amplifiers and label probe according to the manufacturer's instructions (Kit DISCOVERY mRNA, Roche, Cat. #06614353001). Lastly, samples were revealed with DAB staining (3,3′-Diaminobenzidine). The experiment controls used were infected and non-infected HeLa cells.

**Statistical analyses and reproducibility.** All analysis and figures are representative of a single probing experiment. Flow cytometry data were analyzed using FlowJo v10.7.1 software (TreeStar). Data and statistical analyses were performed using Prism 7.0 (GraphPad Software, La Jolla, CA, USA), unless otherwise stated. The statistical specifics of the experiments are provided in the respective figure legends. Data plotted in linear scale were expressed as median + Interquartile (IQR) or Min to Max range, unless otherwise stated. Correlation analyses were performed using Spearman rank correlation. For correlations in Supplementary table 2, a Benjamini–Hochberg False Discovery Rate was also calculated by R Core Team (2020). Mann–Whitney and Wilcoxon tests were applied for unpaired or paired comparisons, respectively, while Kruskal–Wallis rank-sum test with Dunn's post-hoc test was used for multiple comparisons. A P-value <0.05 was considered significant. For most analyses, antigen-specific T cell data have been calculated as the net frequency, where the individual percentage of expression for a given molecule in the control condition (vehicle) has been subtracted from the corresponding SARS-CoV-2-peptide stimulated conditions.

**Reporting summary**. Further information on research design is available in the Nature Research Reporting Summary linked to this article.

## Data availability
The authors declare that the data supporting the findings of this study are available within the paper and its supplementary information files. Source data are provided with this paper.

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

## Acknowledgements

We would like to thank all the patients who participated in the study. We also thank Prof. Shawn C. Kefauver for thoughtful review of the manuscript and Prof. Alex Sanchez-Pla from the Statistics and Bioinformatics Unit (UEB) at the Vall d'Hebron Institut de Recerca (VHIR), for statistical analysis support. This work was primarily supported by a grant from the Health department of the Government of Catalonia (DGRIS 1_5). This work was additionally supported in part by the Spanish Health Institute Carlos III (ISCIII, PI17/01470, and ISCIII COV20/00416), the Spanish Secretariat of Science and Innovation and FEDER funds (grant RTI2018-101082-B-I00 [MINECO/FEDER]), the Spanish AIDS network Red Temática Cooperativa de Investigación en SIDA (RD16/0025/0007), the European Regional Development Fund (ERDF), the Fundació La Marató TV3 (grants 201805-10FMTV3 and 201814-10FMTV3) and the Gilead fellowships GLD19/00084 and GLD18/00008. M.J.B. is supported by the Miguel Servet program funded by the Spanish Health Institute Carlos III (CP17/00179). N.M. is supported by a Ph.D. fellowship from the Vall d'Hebron Institut de Recerca (VHIR) and A.A.-G. and N.S.-G. are supported by a Ph.D. fellowship from the Spanish Secretariat of Science and Innovation (BES-2016-076382, PRE2019-087393). The funders had no role in study design, data collection and analysis, the decision to publish, or preparation of the manuscript.

## Author contributions

Conceptualization, M.J.B. and M.G.; Patient Recruitment and Sample Collection, J.R., A.T., B.P., J.N., P.S., A.L.A., B.A., V.F., J.B.; Methodology, A.F., C.K., J.G.-E., N.S.-G., N.M., M.S., J.E., A.N.A., and S.R.C.; Investigation, J.G.-E., N.S.-G., N.M., M.S., A.A.-G., D.P., D.A.-S., I.S., J.E., A.N.A., and M.G.; Formal Analysis, J.G.-E., N.S.-G., N.M., M.S., D.P., and M.G.; Writing-Original Draft J.G.-E., N.S.-G., N.M., M.S., and M.G; Writing-Review and Editing, R.P.-B., M.J.B., and M.G.; Funding Acquisition, M.J.B. and M.G.; all authors revised the manuscript; Supervision, M.J.B. and M.G. N.M. and M.S. contributed equally.

## Competing interests
The authors declare no competing interests.
