## [Peer Review File · Nature Communications]

REVIEWER COMMENTS

Reviewer #1 (Remarks to the Author):

The study by Grau-Expósito and colleagues examines qualitative and quantitative differences in the T cell response induced in COVID19 patients with varying degrees of disease severity. Major conclusions include: IL-10 is associated with less disease severity and improved outcomes from infection; that a subset of the IL-10+ T cells were more susceptible to cell death, measured indirectly by caspase-3 expression, in hospitalised vs non-hospitalised patients, resident T memory is established after SARS-CoV2 infection.

There is a comprehensive set of data that is analysed using quantitative and qualitative measures. It goes to great lengths to look at function of CD4 and CD8+ T cells responses specific for distinct SARS-CoV2 proteins.

Much of what was described here has been observed in other studies (eg IL-10 linked to disease outcome). The most novel aspect of this study is the observation that TRM can in fact be established after SARS-CoV2 infection. This has been observed in preclinical models but this to my knowledge is the first, albeit preliminary, observation for human infection.

While the experiments are well done, and the conclusions largely supported by the data (except the correlation analyses which I will discuss below), the novelty is somewhat tempered given the vast amount of information already published on T cell responses. This includes descriptions of T cell subsets (TEMRA etc...) outlined in the recent study by Crotty and Sette in Science this year (Dan et al. Science, 2021, 371(6529):eabf4063). More over, a broader dissection of the CD4+ T cell response (particularly TFH, Treg) would have strengthened the overall message of the study (eg does increases in TFH also correlate with better disease outcome as has been reported in other studies; also the role of dysfunctional TFH vs Treg responses and role in disease severity (Meckiff et al., Cell. 2020 183:1340-1353.e16). As a side note, it has been shown that TFH are also more susceptible to apoptosis, an observation that has been proposed to explain the lack of Ab longevity after CoV infection (Kanko et al., Cell. 2020 Oct 1;183(1):143-157.e13)

Finally, I appreciate that the sample size is quite small (something acknowledged by the authors), but this means that some of the conclusions based on the correlation analyses are not fully supported. For example, Fig 3, Fig 4 and Fig 5. The correlation coefficient is shown but the regression line for those r values is missing. This is key to be able to see how the data points line up.

The identification of TRM is not altogether surprising given recent reports for influenza and other respiratory infections. While an interesting and potentially important observation, it would be good to be able to strengthen the link to TRM formation and subsequent protection, something that is difficult at this time given the need to follow up recovered patients for re-infection. As such, the data as it is presented is very preliminary without any insight into the role TRM may play in protection from re-infection. This would be particularly interesting to look at in the context of serologically distinct SARS-CoV2 viruses (eg UK B117 vs SAB1.135) to see if heterologous immunity can be established.

Reviewer #2 (Remarks to the Author):

One of the areas of SARS-CoV-2 T cell immunology which has been most urgently awaited is the characterization of the respiratory response in terms of immunity and immunopathology. Notwithstanding some tasters from RNAseq of patient BAL, all have assumed a major role for lung migration and residence, but little has so far been reported. This study is commendable for taking some first steps in this direction. In fact, it is very much a paper in two parts, Figures 1-5 describing immune parameters in 46 acute patients ranging from mild/non-hospitalized to severe, and then figures 6 and 7 reporting immune analysis of cells from lung biopsy (\pm matched PBMC) from five individuals. The way in which the title and abstract had been presented meant that this

study design took a while to become apparent, which made the paper harder than it should have been to digest. For this reviewer's taste, the Abstract made poor use of the word allocation, barely describing key, novel aspects of the data including the relatively novel cytokine differences observed, and certainly, little idea of which cells from where had been analysed. Some of the wording there is perhaps a little misleading, in terms of 'demonstrate that specific T cells can migrate and establish resident memory' – which is not absolutely the same as showing in one part of the study that specific cells indeed have lung homing properties, CCR7-, and in another part of the study, that specific cells are present in lung biopsies.

Nevertheless, there are many novel and interesting observations here. The authors have elicited noteworthy patterns of differential gamma, IL-4, IL-10 profiles - which merit more attention in the discussion and abstract. The IL-12p70 data are also interesting. There has been much concern about the potential dangers of inducing Th2 lung immunopathology (including at live challenge after vaccination) and this study has some of the most interesting data I've seen about IL-4 profiles, especially the lung response in Figure 7 – perhaps worth a little more appraisal.

There is a typo on page 5 line 26, where 'on' should be 'in.'

The work shown in Figs 6 and 7 from lung biopsy cells is fascinating and novel, including the predictable but important observation that its possible to have lost detectable Ab in serum and T cell response in the periphery, yet have a responsive, polyfunctional TRM population – a clue that correlates of protective immune memory are likely more complex and localised than currently allowed for in simple analyses.

On page 10, line 32, 'contemporaneous' would be a better term than 'contemporary.'

Reviewer #3 (Remarks to the Author):

The manuscript by Grau-Expósito et. al. utilises overlapping peptide pools to assess and functionally phenotype SARS-CoV-2 specific T cells in both peripheral blood during the acute phase of infection, or within lung tissue at varying time points post-viral clearance. Utilizing peptide pools from the different SARS-CoV-2 proteins allows for specific responses against these different proteins, which is important for vaccine design where only individual proteins are employed.

The authors report several findings including that the disease severity of COVID-19 positively correlates with an increased frequency of IL-4-producing CD8 T cells, higher frequencies of TEM (than TCM) and increased T cell apoptosis. Interestingly, the authors also observe that higher frequencies of CD69+CD103+/- T cells that respond to viral antigen, as compared to CD69- T cells, persist within the lung tissue after viral clearance

Major Comments:

- The most interesting part of the paper is the demonstration that SARS-CoV2 specific CD69+CD103+/- TRM are maintained within the lung post infection, which to the best my knowledge hasn't been shown elsewhere. Figure 7 focuses on the cytokine producing function of these cells, which is highly variable across patients, and it was unclear what conclusions were from these analyses. Can the Authors comment.
- Many of the correlation analyses have very minor associations, and it is unclear whether corrections for multiple comparisons has been done (notably SuppTable2). Significant associations could be filtered by employing an adjusted false-discovery rate. It would be nice to see the fitted regression line because for a lot of these associations it is difficult to visualise the relationships often. To determine the most important or impactful relationships, it could be beneficial to perform a dataset wide analysis (Spearman correlation correlogram, with adjusted FDR eg. performed in Mathew et. al. Science, 2020;).

Minor Comments:

- Figure S1, the authors mention CXCL10 being associated with impaired T cell responses, but given CXCL10 strongly correlates with IFN γ (which is elevated), might this not be associated with simply high IFN γ responses.

- Figure 1 could benefit from an analysis where the T cell responses to all peptides (i.e. S, M, and N) are combined. Is it not more physiologically relevant during infection to consider all responses together, rather than separated by protein? In terms of vaccination, where only the spike protein may be used, this becomes more relevant.
- Figure 2/S2A could have benefited from a CD8 stain. CD3+ CD4- cells are not necessarily all CD8+ T cells
- Figure 2B should reference Rha et al., *Immunity*, 2021, who essentially showed the same effect with virus-specific cells (using different TCM/TEM markers). How do these features correlate with viral load?
- Figure 3 could benefit from analysis that combines the data from all peptide pools, given the relatively weak correlations, and it seems clear that every clinical parameter has been correlated to every T cell phenotype measurement. In which case multiple comparison corrections should be applied for all p-values.
- “CD69+ T cells did not express Tbet, confirming their TRM nature” – this is not an accurate statement. Although some CD103+ TRM populations in mice exhibit reduced Tbet (Laidlaw *Immunity* 2014, Mackay *Immunity* 2015). This is not true across TRM subsets, tissues, nor in human samples.
- Again I wonder if a combined analysis of all peptide responses in Figure 6 may reveal differences between TRM/non-TRM groups. I understand the desire to investigate protein-specific responses for questions relating to vaccine efficacy, but for understanding the biological differences between TRM and non-TRM an analysis of the total response may be informative.

REVIEWER COMMENTS

Reviewer #1 (Remarks to the Author):

The study by Grau-Expósito and colleagues examines qualitative and quantitative differences in the T cell response induced in COVID19 patients with varying degrees of disease severity. Major conclusions include: IL-10 is associated with less disease severity and improved outcomes from infection; that a subset of the IL-10+ T cells were more susceptible to cell death, measured indirectly by caspase-3 expression, in hospitalised vs non-hospitalised patients, resident T memory is established after SARS-CoV2 infection.

There is a comprehensive set of data that is analysed using quantitative and qualitative measures. It goes to great lengths to look at function of CD4 and CD8+ T cells responses specific for distinct SARS-CoV2 proteins.

Much of what was described here has been observed in other studies (eg IL-10 linked to disease outcome). The most novel aspect of this study is the observation that TRM can in fact be established after SARS-CoV2 infection. This has been observed in preclinical models but this to my knowledge is the first, albeit preliminary, observation for human infection.

While the experiments are well done, and the conclusions largely supported by the data (except the correlation analyses which I will discuss below), the novelty is somewhat tempered given the vast amount of information already published on T cell responses. This includes descriptions of T cell subsets (TEMRA etc...) outlined in the recent study by Crotty and Sette in Science this year (Dan et al. Science, 2021, 371(6529):eabf4063). Moreover, a broader dissection of the CD4+ T cell response (particularly TFH, Treg) would have strengthened the overall message of the study (eg does increases in TFH also correlate with better disease outcome as has been reported in other studies; also the role of dysfunctional TFH vs Treg responses and role in disease severity (Meckiff et al., Cell. 2020 183:1340-1353.e16). As a side note, it has been shown that TFH are also more susceptible to apoptosis, an observation that has been proposed to explain the lack of Ab longevity after CoV infection (Kanko et al., Cell. 2020 Oct 1;183(1):143-157.e13).

We agree with the reviewer that including an even broader Th analyses, specifically Tregs and Tfh would provide a more complete picture of the different fates of T cell help and functions in relation to disease control. But of course, we made initial choices on which functions we aimed to focus and be able to clearly measure in complex ICS-flow cytometry panels. Thus, we decided to prioritize markers of standard T cell population and function with markers of cell homing, which was the main focus of our study and have been largely understudied.

Finally, I appreciate that the sample size is quite small (something acknowledged by the authors), but this means that some of the conclusions based on the correlation analyses are not fully supported. For example, Fig 3, Fig 4 and Fig 5. The correlation coefficient is shown but the regression line for those r values is missing. This is key to be able to see how the data points line up.

As suggested by the reviewer, we have added now the regression line for all correlations included in the Figures.

The identification of TRM is not altogether surprising given recent reports for influenza and other respiratory infections. While an interesting and potentially important observation, it would be good to be able to strengthen the link to TRM formation and subsequent protection, something that is difficult at this time given the need to follow up recovered patients for re-infection. As such, the data as it is presented is very preliminary without any insight into the role TRM may play in protection from re-infection. This would be particularly interesting to look at in the context of serologically distinct SARS-CoV2 viruses (eg UK B117 vs SAB1.135) to see if heterologous immunity can be established.

We agree with the reviewer on the interest of evaluating the actual protection provided by these TRM. However, and fortunately, there are hardly any reinfections observed in the clinics and the time to perform these type of experiments in a much larger cohort will require years of following-up. Animal models will hopefully enlighten us in this sense.

Reviewer #2 (Remarks to the Author):

One of the areas of SARS-CoV-2 T cell immunology which has been most urgently awaited is the characterization of the respiratory response in terms of immunity and immunopathology. Notwithstanding some tasters from RNAseq of patient BAL, all have assumed a major role for lung migration and residence, but little has so far been reported. This study is commendable for taking some first steps in this direction. In fact, it is very much a paper in two parts, Figures 1-5 describing immune parameters in 46 acute patients ranging from mild/non-hospitalized to severe, and then figures 6 and 7 reporting immune analysis of cells from lung biopsy (\pm matched PBMC) from five individuals. The way in which the title and abstract had been presented meant that this study design took a while to become apparent, which made the paper harder than it should have been to digest. For this reviewer's taste, the Abstract made poor use of the word allocation, barely describing key, novel aspects of the data including the relatively novel cytokine differences observed, and certainly, little idea of which cells from where had been analyzed. Some of the wording there is perhaps a little misleading, in terms of 'demonstrate that specific T cells can migrate and establish resident memory' – which is not absolutely the same as showing in one part of the study that specific cells indeed have lung homing properties, CCR7-, and in another part of the study, that specific cells are present in lung biopsies.

The reviewer is right about the evolution of the paper. We think the paper, as a whole, explains a single story which maybe has been difficult to properly summarize in the abstract. Following the reviewer's advice, we have rewritten the abstract which, although limited by the required number of words, we hope now better reflects our message and summarizes the main findings of the study.

Nevertheless, there are many novel and interesting observations here. The authors have elicited noteworthy patterns of differential gamma, IL-4, IL-10 profiles - which merit more attention in the discussion and abstract. The IL-12p70 data are also interesting. There has been much concern about the potential dangers of inducing Th2 lung immunopathology (including at live challenge after vaccination) and this study has some

of the most interesting data I've seen about IL-4 profiles, especially the lung response in Figure 7 – perhaps worth a little more appraisal.

As suggested by the reviewer, we have highlighted some of these observations in the abstract, discussion and as well as in the results section (page 11 for IL-4 in the lung).

There is a typo on page 5 line 26, where 'on' should be 'in.'

Thanks, the typo has been corrected.

The work shown in Figs 6 and 7 from lung biopsy cells is fascinating and novel, including the predictable but important observation that its possible to have lost detectable Ab in serum and T cell response in the periphery, yet have a responsive, polyfunctional TRM population – a clue that correlates of protective immune memory are likely more complex and localised than currently allowed for in simple analyses.

On page 10, line 32, 'contemporaneous' would be a better term than 'contemporary.'

Thanks, the term has been changed accordingly.

Reviewer #3 (Remarks to the Author):

The manuscript by Grau-Expósito et. al. utilises overlapping peptide pools to assess and functionally phenotype SARS-CoV-2 specific T cells in both peripheral blood during the acute phase of infection, or within lung tissue at varying time points post-viral clearance. Utilizing peptide pools from the different SARS-CoV-2 proteins allows for specific responses against these different proteins, which is important for vaccine design where only individual proteins are employed.

The authors report several findings including that the disease severity of COVID-19 positively correlates with an increased frequency of IL-4-producing CD8 T cells, higher frequencies of TEM (than TCM) and increased T cell apoptosis. Interestingly, the authors also observe that higher frequencies of CD69+CD103+/- T cells that respond to viral antigen, as compared to CD69- T cells, persist within the lung tissue after viral clearance

Major Comments:

• The most interesting part of the paper is the demonstration that SARS-CoV2 specific CD69+CD103+/- TRM are maintained within the lung post infection, which to the best my knowledge hasn't been shown elsewhere. Figure 7 focuses on the cytokine producing function of these cells, which is highly variable across patients, and it was unclear what conclusions were from these analyses. Can the Authors comment.

We agree with the reviewer, on the fact that we did not extracted many conclusions out of Fig.7, beyond the fact that they were highly variable among patients and did not correlate between blood and lung parenchyma. With the new version of the manuscript, we have now included two more patients in these analyses and performed the overall peptide response as requested

by this reviewer, which shows significant differences for IFN γ and clear trends for some other functions (see new Fig.6, 7 and S7 and corresponding results). Thus, main conclusions of this part of the study have been included in the new version of the manuscript.

- **Many of the correlation analyses have very minor associations, and it is unclear whether corrections for multiple comparisons has been done (notably SuppTable2). Significant associations could be filtered by employing an adjusted false-discovery rate. It would be nice to see the fitted regression line because for a lot of these associations it is difficult to visualise the relationships often. To determine the most important or impactful relationships, it could be beneficial to perform a dataset wide analysis (Spearman correlation correlogram, with adjusted FDR eg. performed in Mathew et. al. Science, 2020;).**

Considering the concerns of the reviewer, we have now included in Table S2 the FDR adjusted p values. Further, the fitted regression lines are now included in the graphs where correlations are shown. Last, while including a correlogram could provide an interesting overview of these analyses, we left the original graphs since we think may better show individual correlations that support main results. In any case, our intention with the correlation analyses is just to support the main findings of our study.

Minor Comments:

- **Figure S1, the authors mention CXCL10 being associated with impaired T cell responses, but given CXCL10 strongly correlates with IFN γ (which is elevated), might this not be associated with simply high IFN γ responses.**

We agree with this reviewer with the fact that the more likely cause of increased CXCL10 in hospitalized patients and disease severity is the increase of IFN γ secretion detected in these groups, so we have modified the sentence accordingly (page 5 lines 9-10).

- **Figure 1 could benefit from an analysis where the T cell responses to all peptides (i.e. S, M, and N) are combined. Is it not more physiologically relevant during infection to consider all responses together, rather than separated by protein? In terms of vaccination, where only the spike protein may be used, this becomes more relevant.**

As suggested, we have now included the overall net response considering all peptides (adding up M, N and S) for each function. Graphs showing these data have been included in the Sup. Fig.S2 panel C.

- **Figure 2/S2A could have benefited from a CD8 stain. CD3+ CD4- cells are not necessarily all CD8+ T cells**

We agree with this reviewer, however when we planned the study we considered it was enough to define CD3+ CD4- as mostly CD8+ to favor inclusion of other markers in the flow cytometry panels.

- **Figure 2B should reference Rha et al., Immunity, 2021, who essentially showed the same effect with virus-specific cells (using different TCM/TEM markers). How do these features correlate with viral load?**

We have now included the reference (page 7). In terms of correlation analyses, in our study viral load did not correlate with the frequency of these subsets.

• Figure 3 could benefit from analysis that combines the data from all peptide pools, given the relatively weak correlations, and it seems clear that every clinical parameter has been correlated to every T cell phenotype measurement. In which case multiple comparison corrections should be applied for all p-vals.

We have now included a graph that combines all peptides in one analysis (Fig S3 panel D).

• “CD69+ T cells did not express Tbet, confirming their TRM nature” – this is not an accurate statement. Although some CD103+ TRM populations in mice exhibit reduced Tbet (Laidlaw Immunity 2014, Mackay Immunity 2015). This is not true across TRM subsets, tissues, nor in human samples.

We have modified the previous sentence for “CD69+T cells down-regulated T-bet expression, which has been associated to tissue residency”, as well as changed the reference for “Behr et al. Front Immunol 2018”, which provides a more suited overview on the role of this transcriptional factor in tissue residency (page 10).

• Again I wonder if a combined analysis of all peptide responses in Figure 6 may reveal differences between TRM/non-TRM groups. I understand the desire to investigate protein-specific responses for questions relating to vaccine efficacy, but for understanding the biological differences between TRM and non-TRM an analysis of the total response may be informative.

Thanks for the suggestion. We have now included a graph that combines all peptides in one analysis (Fig 6 panel D).

REVIEWERS' COMMENTS

Reviewer #1 (Remarks to the Author):

The revised manuscript by Grau-Expósito goes some way to address original concerns. As the authors point out, the initial data set is focussed on migratory T cell populations which represents a drilling down into the characterisation of T cell responses to SARS-COV2. The data are robust and largely support the conclusions. The addition of the regression lines allows better determination of what data sets are driving the correlations observed. The report of TRM in the lungs of recovered patients months after infection is a significant finding. The addition of two more patient samples helps strengthen these conclusions.

Reviewer #3 (Remarks to the Author):

The reviewers have adequately addressed my concerns